# Visualization of *Trypanosoma brucei* flagellar pocket collar biogenesis identifies two new cytoskeletal structures

Marie Zelená[1☯], Elina Casas[2☯], Chloé Lambert[2], Nicolas Landrein[2], Denis Dacheux[2,3], Eloïse Bertiaux[2], Kim Ivan Abesamis[4], Gang Dong[4], Vladimir Varga[1], Derrick Roy Robinson[2], Mélanie Bonhivers[2]*

**1** Laboratory of Cell Motility, Institute of Molecular Genetics of the Czech Academy of Sciences, Prague, Czech Republic, **2** Laboratoire de Microbiologie Fondamentale et Pathogénicité (MFP), Université de Bordeaux, CNRS, UMR 5234, Bordeaux, France, **3** Bordeaux INP, Microbiologie Fondamentale et Pathogénicité, UMR 5234, Bordeaux, France, **4** Max Perutz Labs, Vienna Biocenter, Medical University of Vienna, Vienna, Austria

☯ These authors contributed equally.
\* melanie.bonhivers@u-bordeaux.fr

## Abstract

Understanding how cells assemble internal structures is central to cell biology. In *Trypanosoma brucei*, the flagellar pocket (FP) is essential for nutrient uptake, and immune evasion, and its formation depends on a cytoskeletal structure called the flagellar pocket collar (FPC). However, the mechanisms underlying FPC assembly remain poorly understood. In this study, we used cutting-edge ultrastructure expansion microscopy (U-ExM) to investigate FPC biogenesis in *T. brucei*. We mapped the formation of the proximal part of the new microtubule quartet (nMtQ) alongside flagellum growth, providing new insights into its assembly. Additionally, we tracked the localization dynamics of key structural proteins—BILBO1, MORN1, and BILBO2—during the biogenesis of the FPC and the hook complex (HC). Notably, we identified two previously undetected structures: the proFPC and the transient FPC-interconnecting fiber (FPC-IF), both of which appear to play crucial roles in linking and organizing cellular components during cell division. By uncovering these novel aspects of FPC biogenesis, our study significantly advances the understanding of cytoskeletal organization in trypanosomes and opens new avenues for exploring the functional significance of these structures.

## Introduction

The biogenesis of cytoskeletal structures is a fundamental process that shapes eukaryotic cell architecture and function. It is essential for maintaining cell shape, facilitating intracellular transport, driving cell division, and enabling cellular motility. This process involves the synthesis of monomeric proteins, their polymerization into

**Data availability statement:** All relevant data are within the paper and its Supporting Information files.

**Funding:** M.B. was supported by the French National Research Agency (ANR) Structu-Ring [grant ANR-20-CE91-0003] (https://anr.fr/), the Centre National de la Recherche Scientifique (CNRS) (https://www.cnrs.fr/en), and the Université de Bordeaux (https://www.u-bordeaux.fr/en). D.R.R. was supported by the Laboratoire d'Excellence « Alliance Française contre les Maladies Parasitaires » (LabEx ParaFrap) (https://www.labex-parafrap.fr/en/) from the ANR [grant ANR-11-LABX-0024], the CNRS and the Université de Bordeaux. V.V. was supported by the Czech Science Foundation (GA CR) [project 23-07370S] (https://gacr.cz). G.D. was supported by the Austrian Science Fund (FWF) [grant I5960-B2] (https://www.fwf.ac.at/en/). K.I.A. was supported by the Austrian Science Fund (FWF) [grant I5960-B2, PhD fellowship] (https://www.fwf.ac.at/en). E.B. was supported by the LabEx Parafrap [grant ANR-11-LABX-0024], the CNRS, and the Université de Bordeaux. D.D. was supported by the CNRS and the Université de Bordeaux, and Bordeaux INP (https://www.bordeaux-inp.fr/en). C.L. was supported by a PhD student fellowship from [grant ANR-20-CE91-0003]. E.C. was supported by the ANR [grant ANR-20-CE91-0003]. M.Z. was supported by a PhD student fellowship from the Faculty of Science, Charles University, Prague, Czech Republic (https://cuni.cz/UKEN-1.html).

**Competing interests:** The authors have declared that no competing interests exist.

**Abbreviations:** BB, basal body; FAZ, flagellum attachment zone; FP, flagellar pocket; FPC, flagellar pocket collar; FPC-IF, flagellar pocket collar-interconnecting fiber; HC, hook complex; MtQ, microtubule quartet; nBB, new basal body; nFg, new flagellum; nFP, new flagellar pocket; nHC, new hook complex; nMtQ, new microtubule quartet; oBB, old basal body; oFg, old flagellum; oHC, old hook complex; oMtQ, old microtubule quartet; proBB, pro-basal body; proFPC, pro-flagellar pocket collar; U-ExM, ultra-structure expansion microscopy

higher-order assemblies, and the coordinated action of various regulatory proteins and signaling pathways. In many protist parasites, which possess highly organized cytoskeletal architectures, these mechanisms are particularly critical as they transition between hosts, undergoing distinct morphological stages.

Trypanosomatids are flagellated protist parasites responsible for devastating human and animal diseases, including human African sleeping sickness, Nagana, Chagas disease, and leishmaniasis. Trypanosome morphology and biochemistry have revealed a dense subpellicular microtubule cytoskeleton that forms a corset-like structure, restricting vesicular trafficking across most of the cell surface [1]. This tightly cross-linked cortical cytoskeleton precludes processes such as endocytosis, exocytosis, and phagocytosis at sites other than the flagellar pocket (FP), a specialized invagination of the plasma membrane at the base of the flagellum. As such, the FP serves as the sole site for endo- and exocytosis and is essential for nutrient uptake, protein secretion, immune evasion, and cell morphogenesis [2–4]. This architectural constraint likely reflects an evolutionary trade-off: limiting dynamic membrane trafficking elsewhere in order to maintain a highly organized and efficient system centred on the FP. This specialization enables critical adaptations for parasite survival, including hydrodynamic flow-based clearance of immune complexes and rapid turnover of surface proteins [2,3].

The FP is a conserved feature among trypanosomatids, and these parasites have evolved specialized structures within the FP distal region to enhance its function. This region, referred to as the neck, is tightly enclosed around the flagellum by the flagellar pocket collar (FPC), a key cytoskeletal structure [4–6]. While flagellar or ciliary pockets are present in many eukaryotic cells [7], including mammalian primary and motile ciliated cells where they support cilia-associated vesicular trafficking [8], the FPC is a unique cytoskeletal structure exclusive to kinetoplastids. It forms a stable, electron-dense ring that encircles the distal end of the FP and plays an essential role in FP biogenesis, membrane trafficking, cell viability and pathogenicity [4,9]. The FP and FPC are closely associated with the hook complex (HC), a cryptic MORN1 protein containing structure [10] and the proximal part of a distinct set of four microtubules known as the microtubule quartet (MtQ) [5,6]. Originally described in *T. brucei* [5], the MtQ is a conserved feature among kinetoplastids, but its organization and associations can differ substantially. For example, in *Trypanosoma cruzi* and *Paratrypanosoma*, MtQ-related microtubules contribute to the cytostome architecture, and *Leishmania* species exhibit additional cytoplasmic and pocket-associated microtubules [11–14].

In G1-phase *T. brucei* cells, the MtQ arises between the mature and pro-basal bodies of the flagellum, wraps around the FP (this is the proximal part of the MtQ and is the main focus of this research), and extends along the entire length of the cell body parallel to the flagellum. Although its precise functions remain unclear, the MtQ is believed to help define FP architecture and assist in the formation of the new FP during cell division. Importantly, in *T. brucei*, a Polo-like kinase (TbPLK) is one of the earliest proteins to associate with the new forming MtQ during the cell cycle [15] suggesting that it may play a pivotal role in initiating or coordinating the early events of MtQ assembly.

The cytoskeleton-associated HC, located distal and adjacent to the FPC, forms a hook-like structure in close proximity to the FPC (reviewed in [10]). The first identified HC protein, MORN1, is essential in the bloodstream form of *T. brucei*, and its depletion disrupts the endomembrane system, although its molecular function remains unclear [16,17].

BILBO1, a kinetoplastid-specific protein, was the first *T. brucei* FPC protein to be identified [4]. It is a protein that polymerizes both in vivo and in vitro [18,19] serving as a structural backbone of the FPC and interacting with multiple binding partners including TbKINX1B (a basal body kinesin), BILBO2 (an FPC protein), and FPC4 (a MtQ-binding protein) [20–22]. These interactions suggest a strong functional link between BILBO1 and microtubule-based structures such as the MtQ. Notably, BILBO1 also interacts with BHALIN, a protein localized to the HC, reinforcing the tight structural association between the FPC and the HC [23]. To date, BILBO1 remains the only protein proven to be essential for the overall biogenesis of the FPC, FP, and HC, positioning it as a master organizer for these structures [4,22,23].

Tomography studies on *T. brucei* have characterized the morphology of the FPC and its association with cytoskeletal structures, including the MtQ, highlighting their close association [9,24,25]. Importantly, FP formation begins with the emergence of a ridge between the old and new FPs [25], but its biogenesis is dependent on the FPC [4]. However, the precise mechanisms governing FPC formation during the cell cycle remain largely unknown.

In this study, we characterize the *T. brucei* cell cycle-related stages of FPC formation at high resolution using ultrastructure expansion microscopy (U-ExM), revealing that FPC biogenesis occurs de novo yet remains closely associated with the existing FPC and proximal aspect of the MtQ. Additionally, we identify a previously uncharacterized transient filamentous structure, termed the FPC-interconnecting fiber (FPC-IF), which contains several FPC-related proteins. Finally, we explore the role of FPC biogenesis in FP formation and function, providing new insights into the assembly and regulation of these essential cytoskeletal structures.

## Results

### Spef1 tracking and U-ExM reveal sequential MtQ formation and flagellum elongation during the cell cycle

To gain a clearer understanding of MtQ formation, we leveraged the advanced imaging capabilities of U-ExM in *T. brucei* [14,21,26] and combined this approach with the generation of specific cell lines. We first developed a cell line expressing an endogenously tagged $_{HA}$Spef1 protein, which labels the proximal aspect of the MtQ between the basal bodies and the FPC as previously described [27]. U-ExM was then applied to detergent-extracted $_{HA}$Spef1-expresssing cells (cytoskeletons). Co-immunolabelling of Spef1 and tubulin allowed us to visualize the formation of the new MtQ (nMtQ) and track new flagellum (nFg) elongation throughout the cell cycle (Fig 1).

In G1-phase cells, Spef1 was detected on the old MtQ (oMtQ) (Fig 1a). As the cell cycle progressed, Spef1 labelling appeared on the newly forming nMtQ (Fig 1b). This nascent nMtQ was not located within the inter-basal body region but instead positioned adjacent to and on the outer periphery of the pro-basal body (proBB). As the nMtQ extended, it remained continuously labelled by Spef1 and gradually approached the oMtQ (Fig 1c-e).

Consistent with previous observations [25], the nFg subsequently rotates around the old flagellum (oFg) while remaining attached to it, repositioning the nMtQ to a more posterior location within the cell (Fig 1f and 1g). Following this rotation, two new proBBs were clearly observed: one associated with the old basal body and the other with the new basal body. Notably, the nMtQ extended to the oMtQ before the formation of the new transition zone (Fig 1d), indicating that nMtQ formation and extension precede nFg elongation.

### Sequential BILBO1 localization reveals key steps in FPC biogenesis and identifies novel proFPC and FPC-interconnecting structures

Previous electron microscopy and epifluorescence studies using a polyclonal antibody against BILBO1 have localized this protein to the FPC, as well as along the MtQ and basal bodies [4,16,20]. To gain further insights into FPC formation,

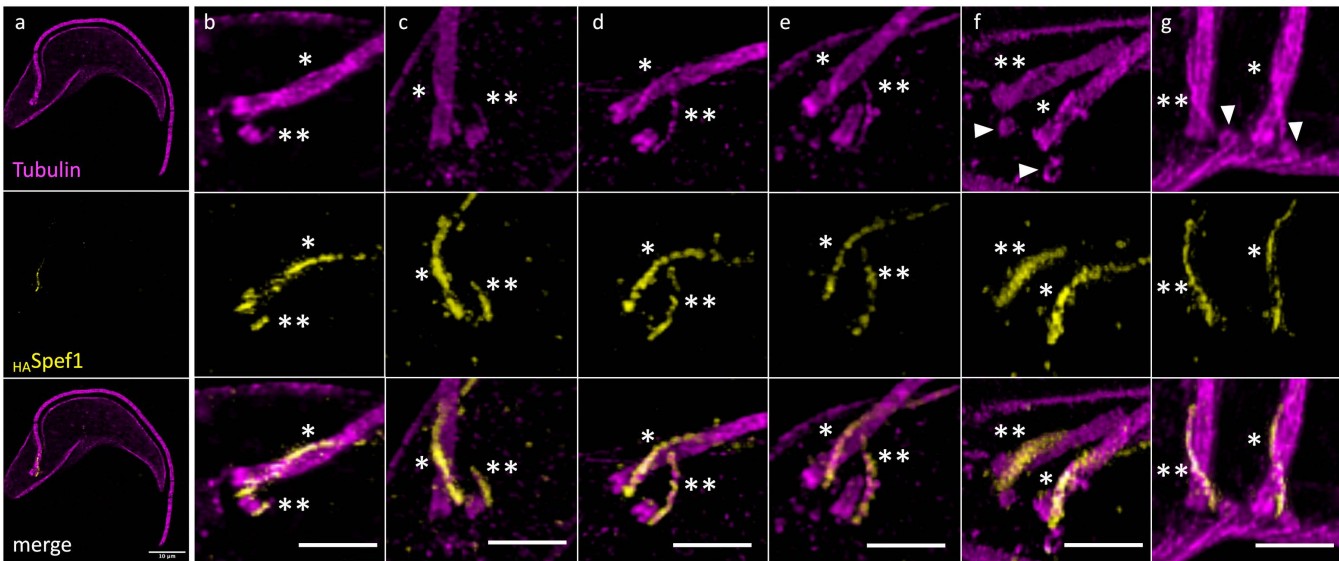

**Fig 1. U-ExM images showing co-immunolabelling of tubulin and HASpef1 on expanded cytoskeletons.** Tubulin (magenta), 10HASpef1 (yellow) (snapshots from 3D viewer). *: old microtubule quartet MtQ. **: new MtQ. Arrowheads: new pro-basal bodies. Scale bars: 10 μm (a), 5 μm (b-g). The data were collected from five independent expanded gels yielding a total of 53 images distributed across cell cycle stages (4 images representative of a, 27 representative of b–e, 22 representative of g).

we used U-ExM to examine BILBO1 localization throughout the cell cycle in relation to Spef1 and tubulin labelling on expanded cytoskeletons (Fig 2).

In early G1-phase cells (Stage 1, Fig 2 Aa), BILBO1 appears as a flattened corkscrew or spring washer-like structure at the FPC. The MtQ, labelled with Spef1, threads through the groove of the FPC (Fig 2 Ab). In addition to the primary FPC-associated signal, weaker BILBO1 labelling is observed distally from the FPC in two linear structures: one along some of the distal aspect of the MtQ (Fig 2) and another corresponding to the mycCAAP1-decorated centrin arm [28] (S1 Fig). As soon as the nMtQ becomes detectable, BILBO1 colocalizes with it, adjacent to the proBB (Stage 2, Fig 2Ba and 2b). Notably, tubulin and BILBO1 co-labelling confirms that the FPC remains tubulin-negative throughout the cell cycle, indicating that tubulin is not a component of the FPC or involved in its biogenesis. Interestingly, at this stage, BILBO1 is absent from the proximal aspect of the oMtQ. However, as the nMtQ extends towards the old FPC, it remains BILBO1-positive (Fig 2Ca), and once the nMtQ reaches close proximity to the old FPC, a BILBO1 signal is also observed on the oMtQ (Fig 2Cb and 2c). The absence of cells in an intermediate state suggests that BILBO1 recruitment to the oMtQ occurs rapidly.

As the nFg grows, a newly formed BILBO1-positive structure emerges, wrapping around the region of the transition zone of the growing flagellum and forming a semi-circular structure that connects the old and new MtQs (Stage 3, Fig 2D). This structure, which we termed the proFPC, serves as the foundation for the formation of the new FPC (nFPC) assembly. At this stage, tubulin labelling reveals two newly formed proBBs (Fig 2Db, arrows).

As the nFg continues to grow and approaches the oFPC, a novel BILBO1-positive, tubulin-negative fibrous structure emerges, linking the old and nFPCs (Stage 4, Fig 2E). This structure, which we termed the FPC-IF, represents also a previously uncharacterized component of FPC biogenesis. The precise mechanism by which BILBO1 is targeted to these structures remains unclear.

As the nFg rotates around the old flagellum before exiting the FP, its distal tip remains attached to the side of the old flagellum *via* the flagellum connector (FC) [25,29]. This rotation, coupled with the movement of the nBB repositions the

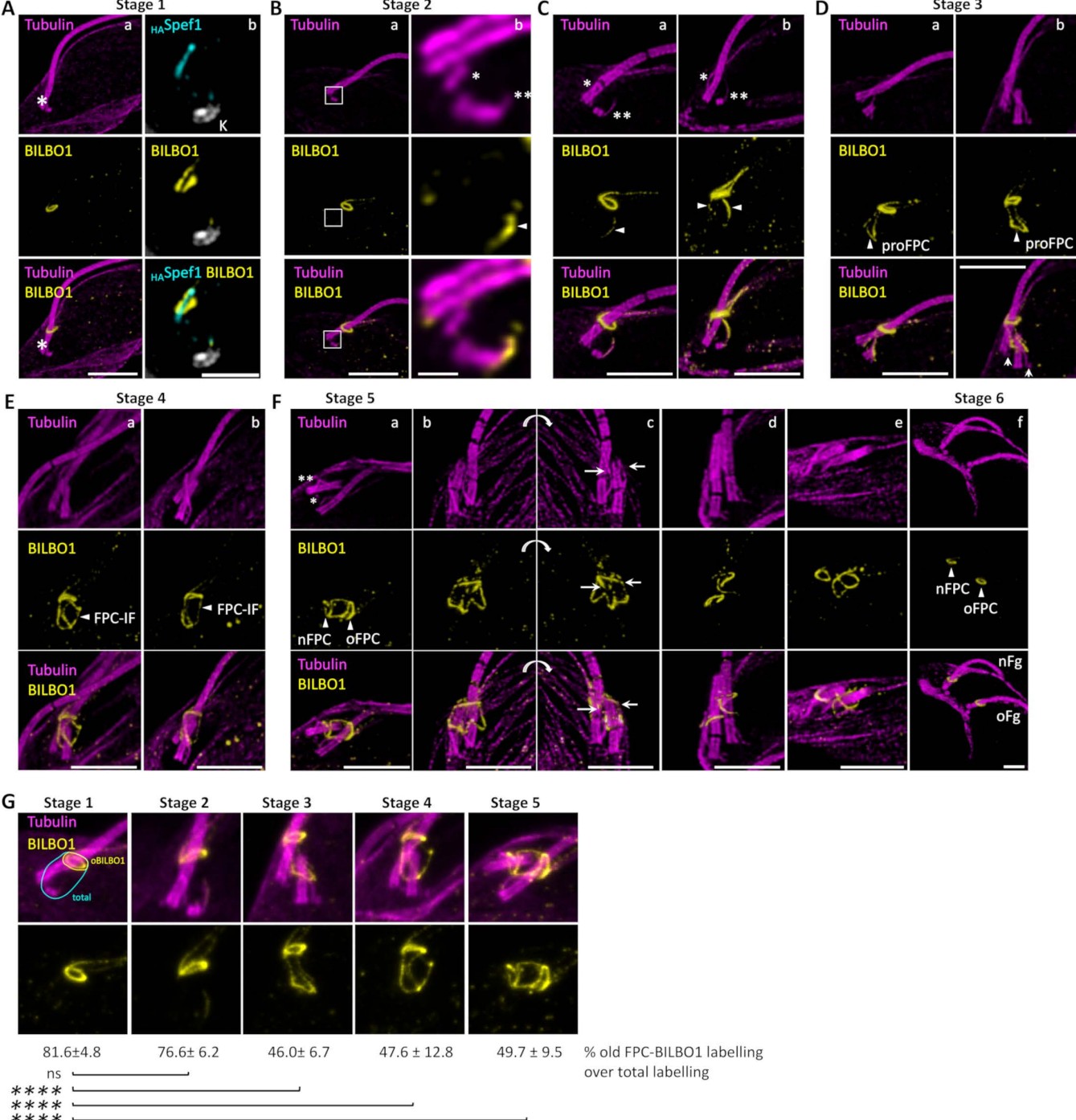

**Fig 2. U-ExM immunofluorescence images showing localization of BILBO1 on expanded cytoskeletons. A.** a. Immunolabeling of tubulin (magenta) and BILBO1 (yellow) labelling. BILBO1 localizes at the flagellar pocket collar (FPC) only. * indicates the microtubule quartet (MtQ). b. Co-labelling of BILBO1 (yellow) and $_{HA}$Spef1 labelling (cyan). A single plane from a Z-stack of Spef1 ($_{HA}$Spef1) and BILBO1 co-labelling reveals that BILBO1 forms a flat corkscrew structure, with Spef1 (and, thus, the proximal MtQ) threading through the corkscrew. The kinetoplast was labelled with Hoechst (K, gray). **B-F**. Co-labelling of tubulin (magenta) and BILBO1 (yellow) at different cell cycle stages. **B.** a. The boxed region is enlarged in b. b. * represents the old MtQ, and ** represents the new growing MtQ. The arrowhead indicates the BILBO1-positive labelling at the new MtQ. **C.** Arrowheads indicate the BILBO1-positive labelling on both the nMtQ and oMtQ. The old MtQ (*) and new MtQ (**) are indicated. **D.** In a, arrowheads indicate the tubulin-negative

pro-FPC structure. In b, arrows indicate the new formed basal bodies. **E.** Arrowheads denote the FPC-interconnecting fibre (FPC-IF). F. Arrowheads indicate the new and old FPCs. nFg: new flagellum. oFg: old flagellum. In b–c, arrows indicate the extremities of the oFPC. **G.** Fluorescence intensity quantification of BILBO1 in representative images, comparing the old FPC (yellow area) to total BILBO1 labelling (cyan area). Quantifications are presented as mean ± standard deviation (SD). **** $p < 0.001$. Number of cells measured (n) was 6, 9, 8, 9, and 7 cells for stage 1, 2, 3, 4, and 5, respectively. The raw measurements underlying the quantitative analyses are presented in S1 Data. Scale bars: 5 μm. The data were collected from 10 independent expanded gel preparation yielding a total of 193 images distributed across cell cycle stages (9 images for stage 1; 47 for stage 2; 38 for stage 3; 38 for stage 4; 40 for stage 5; and 21 for stage 6).

nMtQ to a more posterior location with the cell. Consequently, the nFPC adopts its characteristic corkscrew-like morphology (Stage 5, Fig 2F). Notably, at this stage, the oFPC now adopts an open-circle configuration, creating an enlarged gap (Fig 2Fb and 2Fc). This gap likely serves as the passage of the growing nFg as it exits the oFPC while remaining attached to the old flagellum *via* the FC.

Following BB segregation and subsequent separation of the nFg from the oFg, we observed the disappearance of MtQ-associated BILBO1 labelling and the loss of the FPC-IF. This suggests two key events: (i) cessation of BILBO1 targeting, recruitment or transport (if any) to the FPCs and (ii) the transient nature of the FPC-IF (Stage 6).

To determine whether the quantity of BILBO1 at the oFPC remains stable throughout the cell cycle, we quantified its fluorescence intensity in two distinct regions: one directly surrounding the oFPC (yellow area) and another encompassing the oFPC and extending towards the BB region (cyan area) (Fig 2G).

Quantification of BILBO1 labelling at the oFPC and at the area of nFPC formation (MtQ, proFPC, FPC-IF) reveals that an equivalent amount of BILBO1 is present in the newly forming structures compared to the mature oFPC (Fig 2G). Given that the early stages of nFPC biogenesis occur at the level of the basal body, and that no reduction in BILBO1 is observed at the oFPC, these findings suggest that nFPC formation involves de novo recruitment of BILBO1, rather than redistribution from the existing structure.

## Cytoskeletal dynamics and FPC-IF formation: shaping the flagellar pocket

To investigate the biogenesis of the FP and localize the proFPC and the FPC-IF relative to the FP membrane, we performed immunolabelling with a polyclonal antibody against BILBO1 [16] on expanded whole cells (Fig 3). This is complemented with NHS-Ester Atto 594 labelling of cellular proteins [30] which delineated the FP (purple, Figs 3 and S2c) as an empty space (Figs 3 and S2e-S2g). This method successfully revealed the localization of key FP-associated cytoskeletal structures, including the flagellum, basal bodies, the MtQ, the FPC, and FP boundaries (S2 Fig). We mapped each step of FPC biogenesis, including the formation of the nMtQ (Fig 3Ab), proFPC (Fig 3Ac-Ae), nFg (Fig 3Ac-Ai), FPC-IF (Fig 3Ad and 3Ae), and nFPC (Fig 3Ag-Ai), in relation to the FP.

Whilst the precise timing of FPC-IF formation remains uncertain, our imaging confirmed that the proFPC forms in the cytosolic compartment beneath the base of the FP, and connects the two MtQ structures. Consistent with the previous U-ExM cytoskeletal data (Fig 2), the FPC-IF extends along the anterior side of the FP connecting the proFPC and the oFPC. Following nFg rotation, the FPC-IF disappears (Fig 3Ah), coinciding with a corresponding loss of most BILBO1 labelling on the MtQs (Fig 3Ag). Notably, we observed the relaxation of the oFPC into an open conformation (Fig 3Ai), with this structural transition facilitating nFg exit from the old FPC despite remaining attached to the oFg *via* the FC. Additionally, two distinct FPs were observed as soon as the nFPC was formed (Fig 3Ag).

Measurements in whole cell samples showed that the nFg elongates from stage 2 in a linear fashion (Fig 3Ba) and the distances between the oBB and the oFPC indicate that the mature FPC remains relatively stationary along the oFg during the cell cycle (Fig 3Bb). As previously described in [31], the oBB-nBB distance increases during the cell cycle (Fig 3Bb). In contrast, the distance between the nBB and the proFPC/nFPC reveals that the proFPC forms early in the process of nFg growth (Fig 3Ba and 3Bc). The distance between the nBB and nFPC increases between stage 4 and stage 5, and coincides with the timing of nFg rotation, before stabilizing between stages 5 and

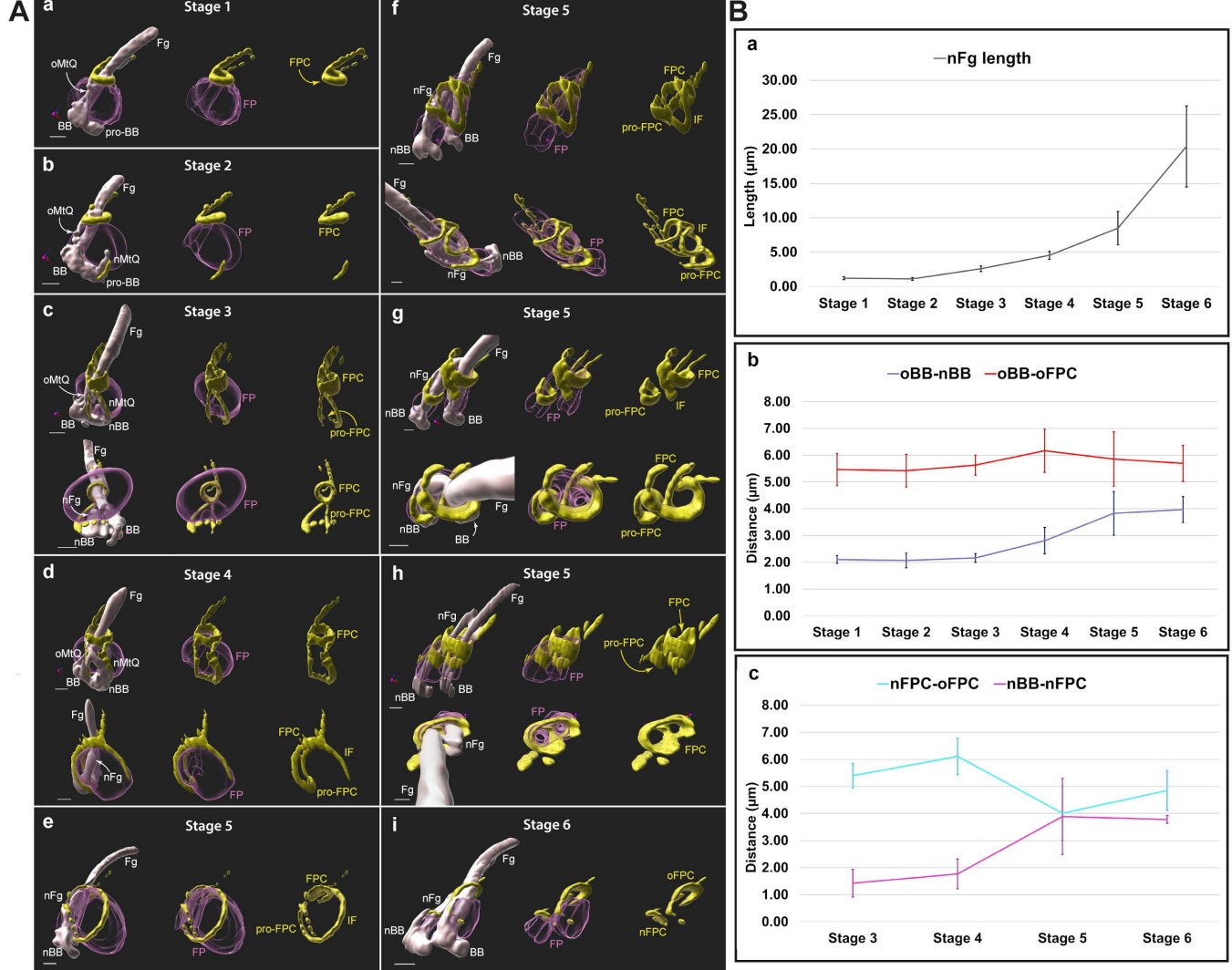

**Fig 3. A. Representation of the flagellar pocket region in successive stages of flagellar pocket division and new flagellar pocket collar formation.** These models were segmented from whole, expanded cells labelled with fluorescent NHS ester (Atto 594) and anti-BILBO1 rabbit polyclonal antibody, followed by an anti-rabbit antibody (Alexa 488). Microtubule-based structures (gray) were segmented from NHS ester data by manual masking and thresholding. The data were collected from seven independent expanded gel preparations yielding a total of 48 images distributed across cell cycle stages as follows: 7 images for stage 1; 5 for stage 2; 8 for stage 3; 5 for stage 4; 11 for stage 5; and 12 for stage 6. Scale bar: 2 μm physical distance (correspond to 0.43 μm after correction for the 4.6-fold expansion factor). Fg—proximal region of the old (mature) flagellum, nFg—new (growing) flagellum, BB—mature basal body belonging to the old flagellum, pro-BB—probasal body, nBB—new mature basal body, oMtQ—old microtubule quartet, and nMtQ—new microtubule quartet. The flagellar pocket (FP, purple) was segmented manually from NHS ester data. BILBO1 antibody signal (yellow) is represented by intensity thresholding. FPC—flagellar pocket collar, pro-FPC—newly forming flagellar pocket collar, oFPC—old flagellar pocket collar, nFPC—new flagellar pocket collar, IF—interconnecting fibre. Progression of FP division: a G1-phase cell: A single flagellum is present before the formation of the nMtQ and nFg. BILBO1 (yellow) forms a loop around the Fg (FPC) and follows a part of the oMtQ distally from the FPC. b Early MtQ formation: Cells with a single flagellum cell were used to collect images of growing nMtQ and show that BILBO1 (yellow) is recruited alongside the growing nMtQ. c. nMtQ reaches the FPC: A nFg starts to assemble, and BILBO1 signal (yellow) is detectable on both the oMtQ and the forming pro-FPC at the base of the nFg. d nFg protrusion: A cell with nFg protruding into the FP, but not yet reaching the FPC. BILBO1 (yellow) forms and interconnecting fibre (IF), which originates from the FPC and follows the surface of the FP. e IF connection to proFPC: A later stage of d, the IF connects to the pro-FPC. f nFg rotation: The growing nFg rotates counter-clockwise around the oFg when viewed from the basal body toward the flagellum. The nBB is now positioned posterior to the BB. The FP starts to divide into two compartments, and the pro-FPC moves distally away from the nBB. g nFg approaches the FPC: The FPC loop relaxes to accommodate the nFg. h nFg reaches the FPC: At this stage, both the Fg and the nFg are enclosed within the FPC. However, in later stages, the nFg is only surrounded by BILBO1 at the nFPC. i nFg extends past the FPC: The nFPC is now fully formed and separates from the original FPC, which is now designated as of oFPC. **B.** Graphs representing measurements of cells in different stages. The measured distances

shown in the graphs are uncorrected by the isotropic ~ 4.6-fold expansion factor. Each plotted value represents the mean of all values measured within each stage ± standard deviation. For each stage, five cells were measured. For the purpose of the measurements, the stages of nFg growth and FP division were divided into six stages: Stage 1: cells before nMtQ formation (represented by 3 Aa), Stage 2: cells with a forming nMtQ, but not yet forming nFg (represented by 3Ab), Stage 3: cells with nMtQ reaching the FPC and the nFg starting to form (represented by 3Ac), Stage 4: cells with nFg with a fully formed transition zone, but not yet reaching the FPC (represented by 3Ad), Stage 5: cells with nFg about to reach or reaching the FPC (represented by 3Ae–h), and Stage 6: cells with nFg past the FPC and fully divided FP before the two FPs move apart (represented by Ai). a—The progress of new flagellum (nFg) growth throughout measured stages. b—Dark blue line: distances between the new basal body (nBB, in some stages pro-BB) and the oBB. The distance was measured from centre to centre from the most distal of each basal body. The distance increases in stages 4-6, suggesting the basal bodies start moving apart in stage 4. Red line: distances between the old basal body (oBB) and the old FPC (oFPC). The distance between the oBB and the nFPC was measured at the furthest point of the nFPC from the oBB. The measurements suggest that the distance between the oBB and oFPC remains constant. c—Cyan line: Distances between the nFPC (or pro-FPC) and the oFPC. The distances were measured between the furthest point of the oFPC relative to the oBB and the apex of the pro-FPC in stages 3 and 4, and between the furthest point of the oFPC relative to the oBB and the furthest point of the nFPC relative to the nBB in stages 5 and 6. The distance between both FPCs decreases between stages 4 and 5. Magenta line: The distance between the nFPC (pro-FPC) and the nBB. The distance between the nFPC and nBB is increased between stages 4 and 5. The raw measurements underlying the quantitative analyses are presented in S1 Data.

6 (Fig 3Bc). These observations suggest three possible scenarios: (i) the proFPC moves upwards along the growing nFg and (ii) the nBB moves downward into the cell. However, it is most likely that both processes occur simultaneously or at specific time points. Indeed, the first scenario is supported by the observed reduction in the distance between the nFPC and oFPC from stage 4 to stage 5, suggesting that the nFPC progressively moves closer to the cell surface (Fig 3Bd). The second scenario is supported by the relatively large change in the nBB-nFPC distance between stage 4 and 5 (Fig 3Bc). Additionally, these observations indicate that the new FP is not fully distinguishable morphologically until the nFPC is completely formed (Fig 3Af), implying that pocket completion depends directly on the formation of the nFPC.

## MORN1 localization during the cell cycle reveals parallel biogenesis pathways for the FPC and the HC

In G1-phase expanded cytoskeletons, MORN1 localized to the bilobe structure [10], which comprises the HC and the centrin arm, positioned adjacent to the distal side of the FPC (Figs 4Aa and S1). This localization pattern is consistent with previous findings [16,21] and closely mirrored that of BILBO1 throughout the cell cycle. In addition to marking the old HC (oHC), MORN1 was detected in expanded cytoskeletons along both MtQs (Fig 4Ab and 4Ac), around the newly forming axoneme (proFPC Stage 3, Fig 4Ad), and later within a filamentous structure connecting to the oHC (FPC-IF Stage 4, Fig 4Ae). This sequential localization pattern culminated in the eventual separation of the old and new HC structures (Fig 4Af). The striking similarity between MORN1 and BILBO1 labelling strongly suggests that MORN1 is consistently present within BILBO1-positive structures throughout the cell cycle, highlighting the functional interplay between the two structures. This hypothesis was supported by observations in a cell line expressing HA-tagged BILBO1 (S3 Fig). Co-labelling experiments, combined with NHS-ester staining, revealed that MORN1 consistently colocalizes with BILBO1-positive structures throughout the cell cycle. However, we detected notable phenotypic alterations in this cell line, indicating that the tag may interfere with BILBO1 function. Given the central role of BILBO1 in FPC architecture, we chose not to pursue further analysis with this cell line to avoid misinterpretation due to potential tagging artefacts.

MORN1 labelling in expanded whole cells (Fig 4B) further emphasized the similarities between the FPC and the HC formation, showing MORN1-decorated MtQs and the emergence of a proFPC-like structure (named proHC in Fig 4Be) linking the two MtQs. This structure subsequently migrates away from the nBB, giving rise to an nHC. However, in these expanded whole cells, MORN1 was not detected on the nMtQ prior to its presence on the oMtQ. This discrepancy is most likely due to differences in labelling sensitivity, as MORN1 is clearly detected on the nMtQ in expanded cytoskeletons (Fig 4A).

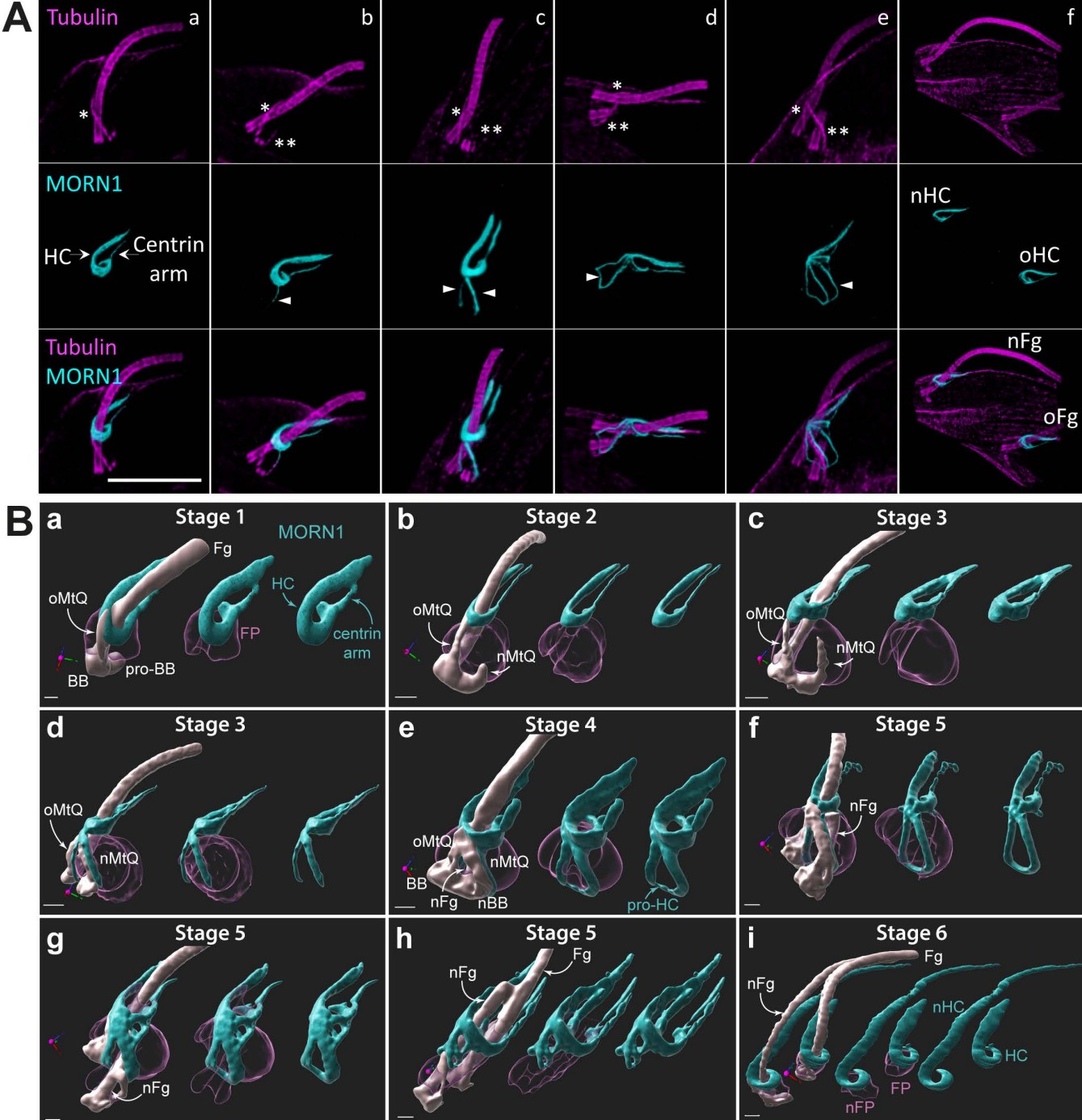

**Fig 4. U-ExM immunofluorescence images depicting the formation of the hook vomplex (HC). A.** Localization of MORN1 on expanded cytoskele-tons that were labelled for tubulin (magenta) and MORN1 (cyan). (a) The HC and the centrin arm of the bilobe structure are indicated by arrows. The old microtubule quartet (MtQ) is marked with a single asterisk. (b) The new growing MtQ is marked with two asterisks, with MORN1 detected exclusively on the growing new MtQ (arrowhead). (c) MORN1 is detected at both the new and the old MtQ (arrowheads). (d) The pro-flagellar pocket collar (proFPC) structure is MORN1 positive (arrowhead). (e) The FPC-interconnecting fibre is also MORN1 positive (arrowhead). (f) At the final stage of the cell cycle, the two flagella separate with the HC and centrin arm. Scale bar: 5 μm. **B.** Segmented views of the flagellar pocket region at sequential stages of fla-gellar pocket division and HC formation in expanded whole cells labelled with anti-MORN1 and NHS ester. The oMtQ is marked with a single asterisk, while the new growing MtQ is marked with two asterisks. The arrowheads indicate the MORN1 labelling on the MtQs (d) and the proFPC-like structure

(e). Scale bar: 2 µm physical size (corresponding to 0.43 µm after correction for the ~4.6-fold expansion factor). Fg—proximal region of the old (mature) flagellum, nFg—new (growing) flagellum, BB—mature basal body belonging to the old flagellum, pro-BB—probasal body, nBB—new mature basal body, oMtQ—old microtubule quartet, nMtQ—new microtubule quartet. The flagellar pocket (FP, purple) was segmented manually from NHS ester data. MORN1 antibody signal (cyan) is represented by intensity thresholding. HC—hook complex, pro-HC—newly forming hook complex, nFP—new flagellar pocket. For Fig 4A, the data were collected from four independent expanded gel preparations yielding a total of 69 images distributed across cell cycle stages as follows: 9 images for stage 1; 8 for stage 2; 12 for stage 3; 15 for stage 4; 12 for stage 5; and 13 for stage 6. For Fig 4B: The data were collected from three independent expanded gel preparations yielding a total of 22 images distributed across cell cycle stages as follows: 5 images for stage 1; 3 for stage 2; 4 for stage 3; 2 for stage 4; 1 for stage 5; and 7 for stage 6.

Notably, MORN1 was never observed at the BB-proximal end of the MtQ in either detergent-extracted cytoskeleton or whole-cell samples. As previously described, MORN1 appeared only when the nMtQ reached the oFPC level, indicating that BILBO1 associates with the MtQs prior to MORN1. Interestingly, Fig 4B panel G clearly shows the formation of a distinct, flask-shaped nFP forming as the distance between the nBB and the proFPC increases. This observation reinforces the idea that not only the FPC is required to maintain the pocket neck in a closed state, but it also actively contributes to FP formation.

### BILBO1 and BILBO2 colocalization reveals the distinct molecular composition of the proFPC and FPC-IF

In this study, we identified three previously uncharacterized cytoskeletal structures, the proFPC, the pro-HC and the FPC-IF. Our analysis revealed that they are tubulin-negative but contain BILBO1 and MORN1. Furthermore, the absence of a Spef1 signal within these structures indicates that they lack this known MtQ-associated protein (Fig 5A).

Further investigations revealed that BILBO2, a direct interacting partner of BILBO1 and a known FPC component [21], colocalizes with BILBO1 along both the new and old MtQ throughout the cell cycle (Fig 5Ba, 5Bb). Notably, BILBO2 was also detected in the proFPC and the FPC-IF, reinforcing that these structures consist of specific, highly organized molecular components rather than transient, amorphous assemblies (Fig 5Bc and 5Bd). These findings underscore the specialized nature of the proFPC and FPC-IF and their important role in the cytoskeletal organization of *T. brucei*.

### BILBO1, MORN1, and BILBO2 localize to a cytoskeletal structure adjacent to the MtQ

To better understand the spatial organization of Spef1 relative to the MtQ, we performed co-immunolabelling of tubulin and Spef1, followed by fluorescence intensity profiling (Fig 6A). The resulting Spef1 intensity profile revealed two distinct peaks flanking the central tubulin peak, suggesting that Spef1 preferentially localizes on specific microtubules within the MtQ. Measurements of the peak pixel intensities of tubulin and Spef1 showed a mean separation of $0.12 \pm 0.015$ µm (mean $\pm$ SEM) (Fig 6A and 6E). Considering an expansion factor of 4.2× and the 25 nm diameter of a microtubule, this corresponds to an in vivo distance of ~28 nm. These results indicate that Spef1 is preferentially enriched along the two distal microtubules of the MtQ, rather than uniformly distributed across all four. In contrast, the mean distance between the peak intensities of tubulin (MtQ) and either BILBO1 or MORN1 was $0.28$ µm $\pm 0.030$ and $0.30$ µm $\pm 0.026$, respectively (Fig 6B, 6C and 6E), corresponding to roughly an in vivo distance of 70 nm. This suggests that BILBO1 and MORN1 are not directly localized on MtQ microtubules. Finally, co-labelling experiments with BILBO1 and BILBO2 revealed a smaller inter-peak distance of $0.07$ µm $\pm 0.012$, or ~16 nm in vivo, suggesting that these proteins co-localize within a distinct structure adjacent to the MtQ.

These findings suggest that BILBO1, MORN1, and BILBO2 positioned between the basal body and the FPC, localize to a cytoskeletal structure or protein complex associated with the MtQ, but are not directly integrated into it. Alternatively, BILBO1 and MORN1 could be confined to a single microtubule within the MtQ, but this hypothesis is inconsistent with the observed Spef1 localization pattern and inter-protein distance measurements. Therefore, the data support the existence of a specialized cytoskeletal structure adjacent to the MtQ that houses these proteins.

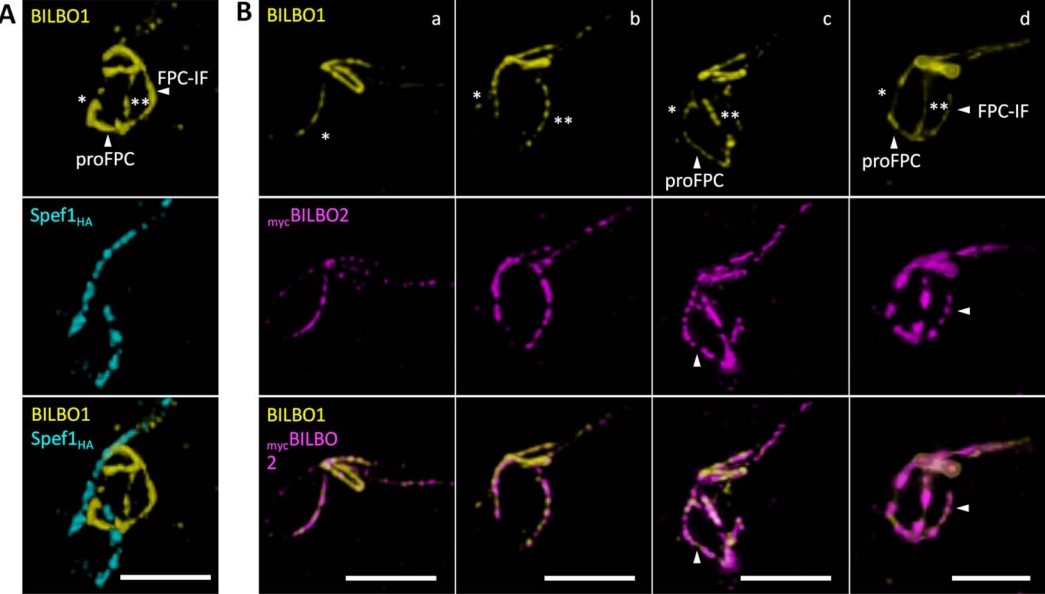

**Fig 5. U-ExM immunofluorescence images on expanded cytoskeletons showing that the pro-flagellar pocket collar (proFPC) and FPC-interconnecting fibre (FPC-IF) both contain FPC-specific proteins. A.** Co-labelling of BILBO1 (yellow) and Spef1 (cyan). **B.** Co-labelling of BILBO1 (yellow) and BILBO2 (magenta). Scale bars: 5 μm.

## BILBO1 knockdown disrupts MtQ orientation but not its assembly

In BILBO1 RNAi-induced knockdown cells, key phenotypes include partial detachment of the nFg at the distal end, the failure to assemble an nFPC, FP, flagellum attachment zone (FAZ), and disorganization of the HC [4,22]. Initial electron microscopy studies of thin sections from BILBO1 RNAi cells did not detect the presence of a newly formed MtQ associated with the nFg and basal body [4,22]. To further investigate whether the formation of the nMtQ depends on BILBO1, we re-examined the BILBO1 RNAi phenotype in expanded samples, using anti-tubulin and anti-MORN1 antibodies. In cytoskeletons of wild-type cells, each flagellum is associated with an MtQ and an HC (Fig 7Aa). However, in BILBO1 RNAi-induced cells, MORN1 and tubulin labelling revealed that the partially detached nFg remains linked to a pro-basal body and a tubulin-positive structure, likely corresponding to the nMtQ (Fig 7B). This finding suggests that MtQ formation can proceed independently of BILBO1 and the FPC.

Interestingly, in BILBO1 RNAi-induced cells (*e.g.,* Fig 7B), the nMtQ and its associated MORN1-positive component exhibited a misoriented trajectory, deviating from the typical alignment along the FP and the FAZ. These observations indicate that while MtQ biogenesis is independent of the nFPC, its correct spatial orientation relies on the FPC. This dependency also extends to the HC structure. Moreover, since no nFPC was formed, it is possible that the proFPC or FPC-IF were either incomplete or absent, suggesting an indirect requirement for BILBO1 in their formation.

Longitudinal electron microscopy sections of WT whole cells, taken from the proximal region of the flagellum (where the central pair is visible), rarely reveal transverse sections of the MtQ. However, in one example (Fig 7Ca), the MtQ appears as an electron-dense region adjacent to the FP membrane. In contrast, in BILBO1 RNAi cells, the absence of an FP correlated with the mislocalization or partial displacement of the MtQ (Fig 7C, 7Cb and 7Cc). These findings further support the idea that BILBO1 is essential for the correct positioning and orientation of the MtQ even though its initial assembly does not require BILBO1.

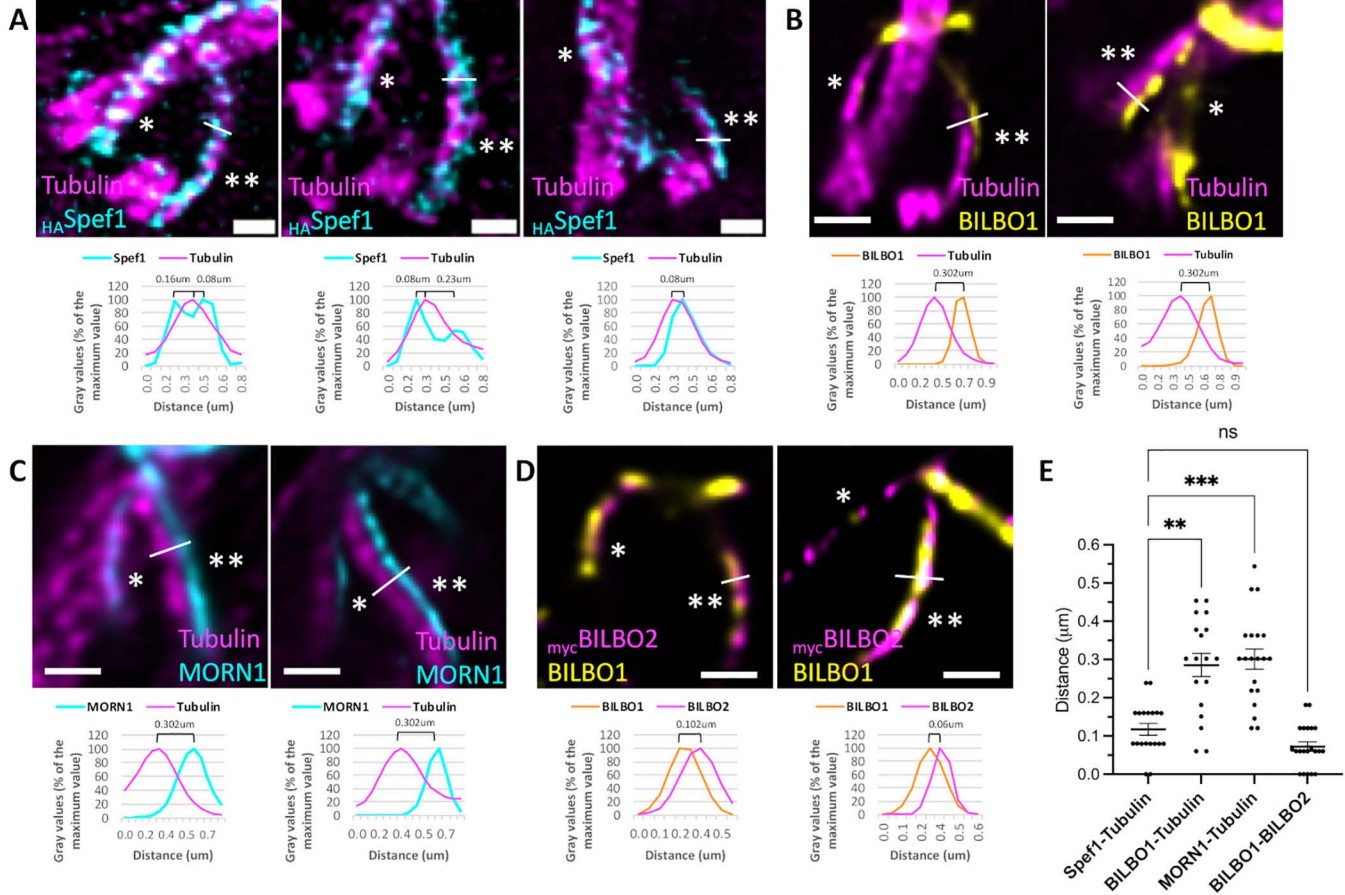

**Fig 6. U-ExM immunofluorescence images on expanded cytoskeletons and intensity peaks measurements showing the spatial relationship between the microtubule quartet (MtQ) (tubulin, magenta) and different markers. A.** Representative images and intensity peaks measurements comparing the MtQ (tubulin, magenta) and Spef1 labelling. **B.** Representative images and intensity peaks measurements comparing the MtQ (tubulin, magenta) and the BILBO1-positive filament. **C.** Representative images and intensity peaks measurements of the MtQ (tubulin, magenta) and the MORN1-positive filament. **D.** Representative images and intensity peaks measurements of the BILBO1-positive filament (yellow) and BILBO2 (magenta). **E.** Statistical comparison of inter-peak distances between markers. Statistical significance was defined as $p < 0.05$, and is indicated on the graph as follows: $p < 0.01$ (**), $p < 0.001$ (***), and non-significant comparison as "ns." The raw measurements underlying the quantitative analyses are presented in S1 Data. Scale bars in A, B, and C: 1 μm.

## Discussion

In this study, we employed U-ExM to investigate the biogenesis of the FPC in *T. brucei*, revealing previously unresolved details of FPC and HC formation. When combined with NHS-ester protein labelling, U-ExM enabled the imaging of novel transient structures involved in FPC/HC assembly and provided the essential cellular context to complement antibody-based protein localization. The high-resolution capabilities of U-ExM offered new insights into the dynamic processes underlying FPC formation. Using these techniques, we uncovered critical interdependencies between the FPC, HC, and MtQ. These findings redefine the molecular composition, functional roles, and dynamic processes of FPC and FP biogenesis.

### FPC biogenesis occurs de novo

While the spatial coordination of events leading to FP formation and division was elegantly illustrated in previous work [32], the specific mechanisms of FPC biogenesis and function remained elusive. As schematized in Fig 8, our findings

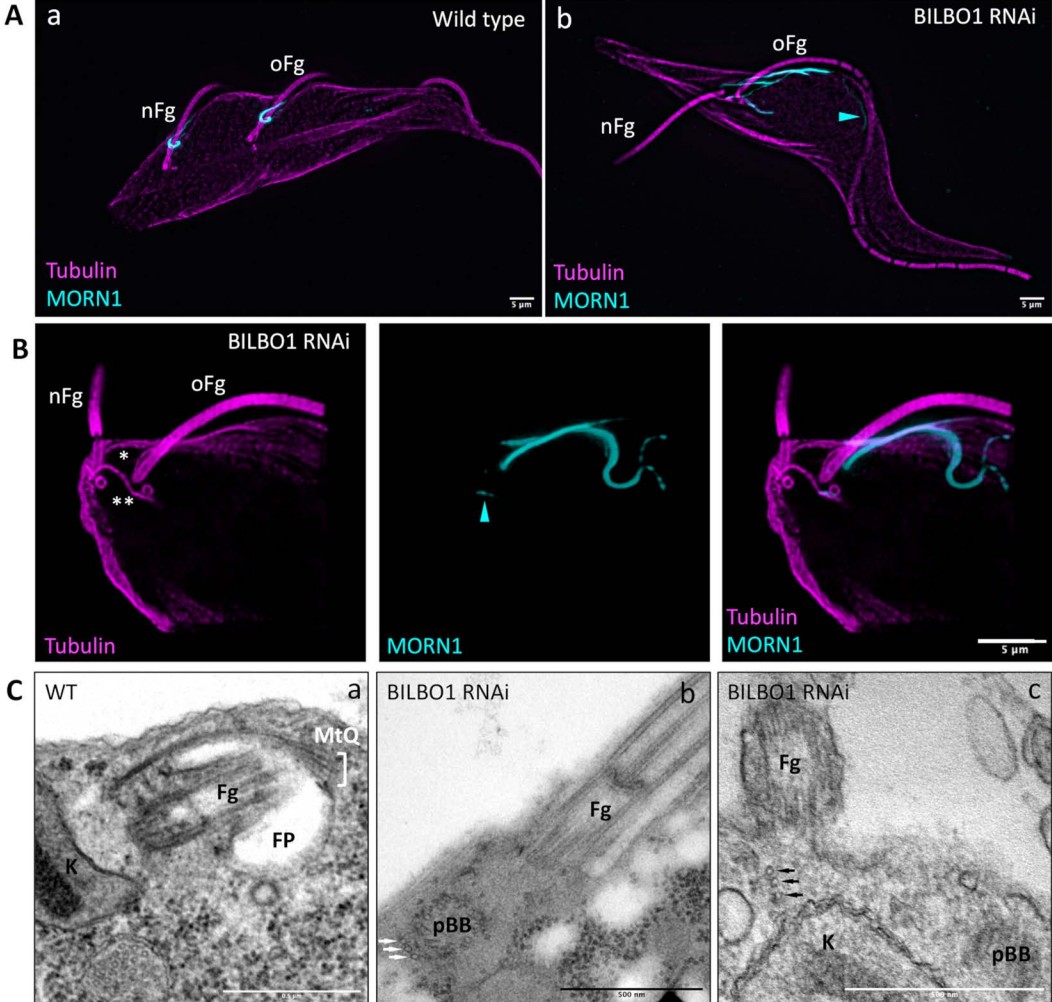

**Fig 7. MORN1 (yellow) and tubulin (magenta) labelling on WT cells and BILBO1 RNAi-induced cells. A.** Epifluorescence acquisition images and Z-project Max intensity representations of (a) expanded cytoskeletons of WT cells showing the hook shape of MORN1 labelling on both the old flagellum (oFg) and the new flagellum (nFg) and of (b) expanded cytoskeletons of BILBO1 RNAi-induced cell displaying disrupted MORN1 localization and detachment of the new flagellum. Scale bars: 5 μm. **B.** Confocal acquisition of an expanded cytoskeleton of BILBO1 RNAi-induced cell exhibiting an advanced phenotype, where the new flagellum is detached along the cell's length. A newly formed microtubule quartet (MtQ) (**) is observed near the basal bodies of the new detached flagellum but has an atypical orientation. MORN1 labelling is also abnormal at the old flagellum, with weak labelling along the nMtQ (arrowhead). Scale bar: 5 μm. **C.** Electron micrographs of thin sections of whole WT cells (a) and BILBO1-RNAi induced cells (b, c). Arrows illustrate sets of intracellular microtubules resembling the MtQ but with three microtubules instead of four. Fg: flagellum; K: kinetoplast; proBB: pro-Basal body; FP: flagellar pocket. Scale bars: 500 nm.

refine their model, demonstrating that FPC biogenesis occurs de novo, initiating at the pro-basal body (proBB) beneath the FP membrane. A nascent MtQ (nMtQ) extends along the proBB. Notably, previous work demonstrated that the nBB and nMtQ are decorated with BILBO1, even though the transition fiber protein TFK1 is not yet detected at the proBB [20,33]. This suggests that nMtQ formation precedes proBB docking, or both events occur concomitantly. As the nMtQ extends and wraps around the FP, it unites with the old FPC. Subsequently, a BILBO1-MORN1-BILBO2-positive filament (the FPC-IF) forms de novo, linking the proFPC to the oFPC at the anterior side of the FP. Just before or during pro-basal/BB body docking at the FP membrane, the oMtQ also becomes decorated with these three proteins. Both MtQs are then

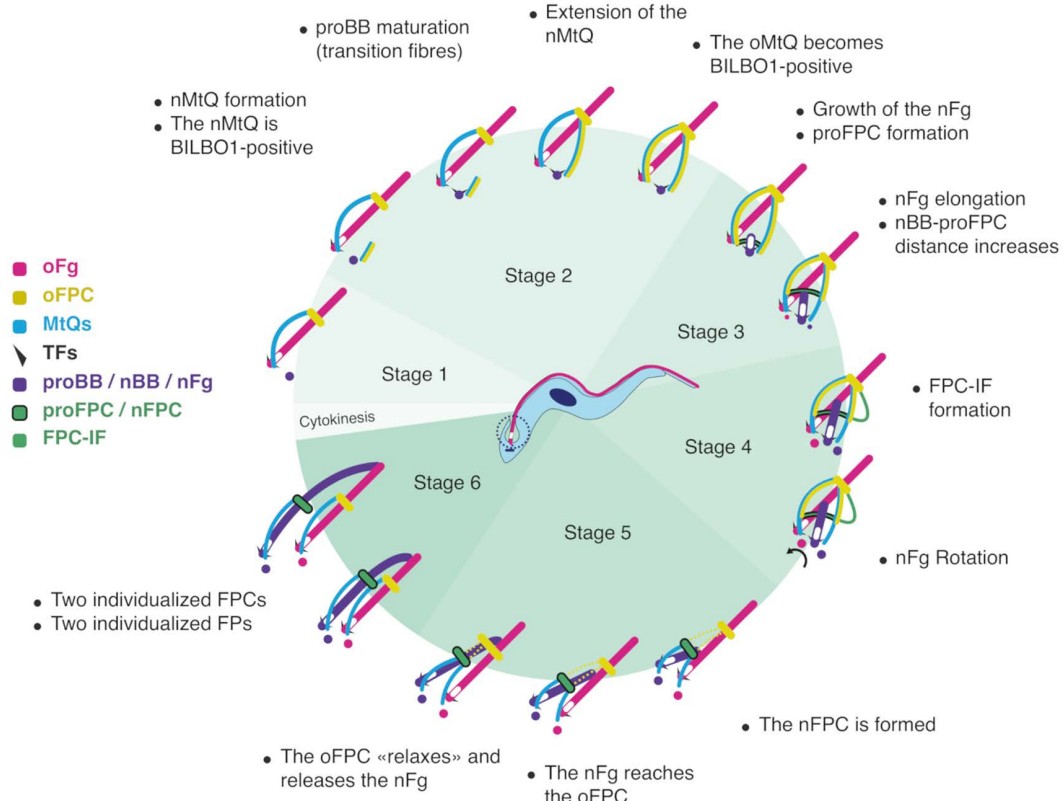

**Fig 8. Schematic model of the flagellar pocket collar (FPC) biogenesis during the cell cycle of *Trypanosoma brucei*.** This diagram illustrates the sequential formation and maturation of the new FPC (nFPC) in six stages, based on ultrastructure expansion microscopy data. Key cytoskeletal components and intermediate structures involved in FPC assembly are indicated. **Stage 1** (G1 phase): Cells possess a single flagellum (old flagellum, oFG, magenta), a single FPC (oFPC, yellow), and a mature microtubule quartet (oMtQ, cyan). No new structures are detectable at this stage. **Stage 2**: The new MtQ (nMtQ, cyan) begins to extend from the vicinity of the pro-basal body (proBB, purple) that matures through the formation of the transition fibres (black triangles), initiating new FPC (nFPC) biogenesis. BILBO1 (yellow) is recruited to the nMtQ early, followed by its recruitment to the oMtQ. This suggests an early role for BILBO1 in scaffolding the developing collar. **Stage 3**: The new flagellum (nFg, purple) begins elongating from the proBB. Simultaneously, the pro-FPC (proFPC, green and black), an immature FPC precursor, appears at the base of the nFg, marked by BILBO1 recruitment. This structure anchors to the nMtQ and the oMtQ. BILBO2 and MORN1also localize to this structure, although they are not depicted in the schematic for clarity. At this stage, new probasal bodies (proBBs, magenta for the proBB associated with the oFg, purple for the proBB associated with the nFg) are observed in association with both the oFg and the nFg, marking an important transition point in the basal body duplication cycle. **Stage 4**: As the nFg elongates and approaches the oFg. At this stage, a novel structure, the FPC-interconnecting fibre (FPC-IF, green) appears, physically linking the oFPC and the proFPC. The nFg basal body undergoes rotational movement relative to the oFg basal body, indicating rearrangement of the cytoskeletal and membrane architecture. **Stage 5**: the new flagellar pocket collar (nFPC, black and green) completes its maturation, forming a helical or corkscrew-like structure. The oFPC undergoes a relaxation or opening event that allows the nFg to exit and emerge from its own fully formed flagellar pocket (not depicted in the schematic for clarity), independent from the oFPC. The disassembly of the FPC-IF likely facilitates this separation, ensuring proper positioning of the nFPC and associated cytoskeletal components for subsequent cell division. **Stage 6**: The two mature FPCs and associated flagella are now fully segregated and associated with individualized FPs. The cell prepares for cytokinesis, with both daughter cells inheriting a complete FP-FPC complex. Note: To simplify the representation, the microtubule quartets anterior to the FPCs and the flagellar pockets are not depicted. Standardized terminology used here: oFPC (old FPC), nFPC (new FPC), proFPC (precursor of the nFPC structure), FPC-IF (FPC-interconnecting fibre), and oFG/nFg (old/new flagellum).

linked *via* the semi-circular proFPC structure, which originates near the nBB transition fibers beneath the FP and serves as the foundation for the nFPC.

As the nFg elongates within the FP, the distance between its new basal body and the proFPC increases, while the basal body undergoes rotary movement towards a more posterior cell position. While the precise mechanism underlying interbasal body movements remains elusive, our data suggest a dynamic interplay between flagellum elongation and

FPC remodeling. Several possibilities exist: (i) the proFPC moves along the flagellum, contributing to ridge and new FP formation, (ii) the proFPC remains stationary while the nFg elongates, or (iii) a combination of both, where the proFPC initially stays in place but later moves along the flagellum as it extends. Our data support the third hypothesis, in which the proFPC remains static at first but later migrates as the flagellum elongates. When the nFg approaches the oFPC, the FPC-IF links the proFPC and oFPC. As the nFg rotates posteriorly, the proFPC closes into a corkscrew-like structure to form the mature nFPC. The FPC-IF then rapidly disappears, although whether this occurs *via* coordinated disassembly or degradation remains unknown. This disappearance marks the final separation of the two FPC structures. We observed a shift in the chronology of proFPC movement: in cytoskeletons it occurs from stages 3–4 (Fig 2), whereas in whole cells, it is delayed until stages 4–5 (Fig 3). We propose that this discrepancy represents an artefact of detergent extraction, arising from the absence of the FP membrane, which in whole cells may normally provide slight resistance to proFPC displacement.

Importantly, we show that the oFPC relaxes into an open circular structure, allowing the nFg to exit the oFPC while its distal tip remains attached to the oFg *via* the FC, before restructuring into its compact, corkscrew-like conformation. This requires precise protein coordination, although the exact mechanism remains unknown. These findings align with research from the Keith Gull lab, supporting the hypothesis that FPC biogenesis occurs de novo [34].

### MtQ biogenesis is independent of the FPC and required for flagellum attachment

We observed that nMtQ formation and extension occur before significant flagellum elongation, suggesting that MtQ biogenesis precedes flagellum growth. This is in agreement with previous findings on early cytoskeletal formation in trypanosomes [25]. In BILBO1 RNAi knockdown cells, proFPC and FPC-IF formation is disrupted, yet the nMtQ still forms near the pro-basal body. This suggests that MtQ biogenesis is independent of the FPC, HC, and FP but relies on an alternative, unidentified mechanism. Furthermore, it is unclear if there is a link with potential transport along the entire MtQ for supporting FAZ assembly and its ER attachment, but the absence of the FAZ and disturbed ER plus the misorientated MtQ in induced BILBO1 RNAi knockdown cells ([4], and this work) is in line with the potential role of the MtQ in providing a platform for transport of FAZ constituents and ER-attachment.

While FPC and HC proteins localize to the MtQ, proFPC, and FPC-IF, the precise mechanism by which they are trafficked to these structures remains unclear. Proteins may be recruited directly from the cytoplasm or are transported along the microtubules by motors. However, several kinesins are positioned on the proximal MtQ, [20,35], and given the theoretical polarity of the MtQ, with microtubule plus-ends orientated away from basal body, their positioning is consistent with a role in anterograde transport from the basal body region towards the FPC.

### Nature of BILBO1-positive structures and insights into HC and FP formation

Our study reveals that BILBO1 forms or contributes to a polymer running parallel to the MtQs and associating with the proFPC and FPC-IF. The polymerization properties of BILBO1, demonstrated both in vivo and in vitro [18,19], along with its interaction with proteins such as BILBO2 [20–23], suggest that BILBO1 may be a key candidate for polymer formation along the MtQs and for de novo structures such as proFPC and FPC-IF.

Our findings also suggest that the BILBO1-MORN1-BILBO2-positive structure at the nMtQ most probably corresponds to the previously described "tendril" [16], which facilitates FPC biogenesis, parallels the nMtQ, and contains a dynamic pool of MORN1. We propose that BILBO1 provides a molecular skeleton for protein recruitment during FPC biogenesis [18,20,21,23] consistent with BILBO1 RNAi knockdowns showing disrupted nMtQ orientation, impaired recruitment of FAZ and interstitial zone layer proteins [36], and mislocalized HC components [23,37]. These findings highlight BILBO1 as a central organizer of multiple FP-associated structures. Our model aligns with previous studies [38], showing that FP formation precedes flagellum separation during proventricular stage development. It is unclear if the proFPC is involved in ridge formation in the early stages of pocket division. However, we propose that

BILBO1, *via* the proFPC, is directly involved in ridge formation after nFg rotation, but the precise mechanisms and driving forces remain to be elucidated. However, in procyclic forms, the FC, an electron-dense structure linking the tip of the nFg to the lateral aspect of the old flagellum, has been proposed to transmit forces that guide basal body separation and flagellar elongation [29]. The timing of these events is notable: the early rotation and limited anterior displacement of the new basal body and associated structures occur while the FC is still actively migrating. This suggests the FC may contribute to positioning during early proFPC assembly. However, the more pronounced posteriorward migration of the nBB typically occurs after FC migration has ceased [39], indicating that the FC may not be the primary force-generating mechanisms throughout the entire process. In bloodstream forms, which lack a flagella connector, a distinct transmembrane structure called the groove has been described at the tip of the nFg [40]. While this may serve as an alternative structural interface or impose mechanical constraints, its formation appears to occur only after the nFg has exited the FP. Therefore, its role in early BB and proFPC positioning remains uncertain. Importantly, studies in *IFT20* and *DHC1b* RNAi procyclic cells, in which new axoneme assembly is impaired, demonstrate that BB rotation and separation can still occur in the absence of a new axoneme [39]. These findings argue against a model in which the FC exerts force exclusively *via* the axoneme. Instead, we propose that early proFPC dynamics may rely on other mechanisms, potentially involving BB-associated motor proteins, local membrane remodeling, or cytoskeletal interactions.

## Unresolved questions in FPC and HC formation

The mechanisms driving the proFPC semicircle structure initiation and the processes that generate the template linking the new and oMtQs remain unclear. BILBO1, which interacts with multiple proteins [20–23], may recruit factors essential for membrane docking and FPC-IF formation. Additionally, it may associate with motor proteins that drive proFPC movement along the axoneme, contributing to new FP formation. Notably, several kinesin were localized at the proximal MtQ [28,35] including TbKINX1B [20]. While TbKINX1B knockdown did not prevent FPC biogenesis in cultured procyclic cells, it caused FPC reorientation to the anterior side of the nucleus, suggesting a role in FPC positioning. In bloodstream forms, TbKINX1B is essential, and its knockdown resulted in formation of enlarged FPs, further supporting an FP/FPC-related function. The exact trigger for proFPC movement remains unknown, but this sliding mechanism likely involves motor proteins that position the maturing FPC closer to the cell surface. This repositioning may facilitate ridge formation, a key step in FP formation.

The transient FPC-IF is described in our study as a novel, short-lived structure that appears to link specific cellular components (the oFPC, the nFPC, the FP membrane) during the biogenesis of the nFPC in *T. brucei*. Its rapid and stage-specific appearance suggests that it may function as a dynamic organizational scaffold, coordinating the spatial positioning of cytoskeletal elements and membrane domains during the emergence of the nFg and the maturation of the new FP. Given the central role of the FP in endocytosis and immune evasion in *T. brucei*, the transient nature of the FPC-IF may reflect a tightly regulated mechanism for preserving membrane compartmentalization and trafficking fidelity during division. Although direct functional evidence remains limited, the transient FPC-IF likely plays a critical role in coordinating the morphogenesis of the FPC in *T. brucei*. By transiently linking the old and newly forming FPCs, the FPC-IF may act as a mechanical and spatial scaffold that ensures proper positioning and segregation of key cytoskeletal components such as BILBO1 and MORN1. This linkage likely stabilizes the FP neck region during the major membrane and cytoskeletal remodeling events that accompany flagellum duplication and FP formation. The dynamic assembly of the FPC-IF may also facilitate the targeting or stabilization of membrane-associated proteins, contributing to the establishment of specialized membrane domains within the FP that are critical for its endocytic and exocytic function. As the FP serves as the exclusive site of endocytosis and exocytosis in *T. brucei*, including for surface glycoprotein trafficking, maintaining its integrity is essential for immune evasion. Additionally, disruption of FP biogenesis, *via* incorrect FPC-IF function or positioning, could compromise endocytic efficiency and immune evasion by mislocalizing surface proteins. Moreover, the

precise but transient appearance of the FPC-IF at the interface of the oFPC and nFPCs suggests a role in shaping membrane curvature and compartmentalization during FP morphogenesis. Collectively, these observations support the notion that the FPC-IF integrates cytoskeletal organization, membrane remodeling, and trafficking to ensure proper FP inheritance and functionality. The disassembly of the FPC-IF likely facilitates proper positioning of the nFPC and associated cytoskeletal components for subsequent cell division. We further speculate that analogous structures may exist in other kinetoplastids, such as *T. cruzi* and *Leishmania*, where conserved FP/FPC architecture suggests the potential for similar mechanisms.

We suggest that the regulatory mechanisms governing proFPC and FPC-IF formation are orchestrated by a cascade of spatially and temporally controlled protein–protein interactions. A central player in this process is BILBO1, which acts as a master organizer of FPC assembly and is recruited early to nascent sites of FPC biogenesis. Its early presence at the proFPC and nMtQ suggests that it may serve as the initial structural seed for complex assembly. This may relate to BILBO1's intrinsic ability to self-polymerize into distinct morphologies, either linear (*e.g.*, FPC-IF) or circular (*e.g.*, mature FPC), depending on local cues. Spatial regulation of BILBO1 polymerization may be modulated by calcium or post-translational modifications such as phosphorylation [18]. The initiation of proFPC assembly likely depends on the spatial definition of nucleation sites, near the base of the forming nMtQ. Subsequent extension along the oMtQ and around the nascent FP neck may reflect guided polymerization. Additional components, such as TbKINX1B and FPC-associated proteins [20–23], may be recruited through direct interactions with BILBO1 polymers and contribute to anchoring or remodeling. Cell cycle–regulated kinases such as NRKC (involved in BB separation) [41] and TbPLK [15] (which localizes to the nMtQ early in the cycle) may further orchestrate this process by phosphorylating scaffolding or adaptor proteins that act as docking platforms. Disassembly of transient structures such as the FPC-IF may be driven by active depolymerization or destabilization events, mimicking mechanisms observed for other cytoskeletal assemblies.

## Conclusions

This study provides key insights into the biogenesis of the key structures FPC, MtQ, and flagellum in trypanosomes. We show that while MtQ formation occurs independently of BILBO1, its proper orientation depends on an intact FPC, underscoring their interplay. Further investigation into the molecular mechanisms governing MtQ orientation and the role of the FPC-IF in FPC formation will enhance our understanding of FP and FPC biogenesis.

## Materials and methods

### Cell lines, culture, and transfection

Procyclic (PCF) cell lines SmOxP427 and SmOxP927 co-expressing the T7 RNA polymerase and tetracycline repressor [42] were grown at 27°C in SDM-79 medium supplemented with 2–2.5 mg/mL hemin, 26 µM sodium bicarbonate, 10% (v/v) complement-deactivated FBS, and 1 µg/mL puromycin. For transfection, cells were grown to $5 \times 10^6$–$1.0 \times 10^7$ cells. mL$^{-1}$, then $3 \times 10^7$ cells were electroporated using transfection buffer as previously described [43,44] with 10 µg of PCR product using the program X-001 of the Nucleofector II, AMAXA apparatus (Biosystems), or $1 \times 10^7$ cells were electroporated with 45 µL of PCR product using the BTX ECM 830 electroporator (3 pulses, 2,000 V, length 100 µs, interval 100 ms, unipolar). After transfection, clones were obtained by serial dilution and maintained in logarithmic phase growth. Selection antibiotics were added to the media according to the transfected product (neomycin 10 µg/mL, hygromycin 25 µg/mL, phleomycin 5 µg/mL, and blasticidin 20 µg/mL).

The BILBO1 stem-loop RNAi vector p3960SL was previously described in [4]. BILBO1 RNAi was induced for 48H with 10 µg/mL tetracycline. Endogenous 10 × Myc-Nter tagged BILBO2 cell line was described in [21]. Spef1 (10xHA tag at its N-terminus or C-terminus) Centrin arm associated protein 1 (CAAP1) (10xmyc at its N-terminus), and BILBO1 (3xHA at its C-terminus) were endo-tagged using the pPOTv7 vector as described in [45].

### Immunofluorescence

**Wide-field and epifluorescence microscopy.** Cells were harvested, washed in PBS, and processed for immunolabelling on methanol-fixed detergent-extracted cells (cytoskeleton, CSK) as described in [22]. The antibodies used and their dilutions are listed in S1 Table. Images were acquired on a Zeiss Imager Z1 microscope, using a Photometrics Coolsnap HQ2 camera, with a 100×Zeiss objective (NA 1.4) using Metamorph software (Molecular Devices), and processed with ImageJ (NIH). The U-ExM wide-field images were deblurred with Metamorph Deconvolution 2D.

**Ultrastructure-expansion microscopy. U-ExM on detergent-extracted cytoskeletons.** The U-ExM protocol has been adapted from [46]. Briefly, $4 \times 10^6$ mid-log phase cells were washed in PBS and loaded onto poly-L-lysine 0.1% solution -coated 12 mm coverslips in 24-well plates and cells left to adhere for 5–10 min. Cells were extracted for a few seconds with PBS NP40 (Igepal) 0.25%. Cells were then covered with 1 mL of activation solution (0.7% formaldehyde, 1% Acrylamide in PBS) for 3.5 h at 37°C. For the gelation step, the coverslips were gently deposited on top of a 35 μl drop of MS solution (23% sodium acrylate, 10% acrylamide, 0.1% bisacrylamide 0.5% TEMED and 0.5% ammonium persulfate in PBS) for 5 min then transferred at 37°C and incubated for 1 h in a moist chamber. The coverslips were then transferred to a 6-well plate in 1 mL of denaturation solution (200 mM SDS, 200 mM sodium chloride, 50 mM Tris-HCl pH 9.0) with agitation at RT for 15–30 min to detach the gel from the coverslip. The gel was then moved into a 1.5 mL Eppendorf centrifuge tube filled with denaturation solution and incubated at 95°C for 90 min. Gels were expanded in 100mL of deionized water (3×30 min) then incubated 3×10 min in 100mL of PBS. Small pieces of the gels were processed for immunolabelling as follows. The gels were preincubated in blocking solution (PBS, 1%–2% BSA and 0.2% Tween20 or PBS only) for 30 min at 37°C, and then with the primary antibodies (see S1 Table for the reference and dilutions) overnight at 37°C with shaking. After three washes in 1 mL blocking solution, the gels were incubated with the secondary antibodies diluted in blocking solution for 4.5 h in the dark at 37°C with slow agitation. After three washes in blocking solution, gels were expanded in 100 mL of deionized water (3×30 min). An expansion factor was determined using the ratio between the size of the coverslip (12 mm) and the size of the gels after the first expansion and was 4.2-fold.

**U-ExM on whole cells.** The U-ExM protocol on whole cells was performed as previously described in [14]. Briefly, $2 \times 10^6$ cells were washed 2x in PBS, fixed with 4% formaldehyde and 4% acrylamide in PBS, seeded onto 12 mm round coverslips washed in NaOH, and incubated overnight. Next day, the coverslips were gently washed in PBS and inverted cell side down onto a 50 μL drop of polymerizing monomer solution (19% sodium acrylate, 10% acrylamide, 0.1% N,N'-methylenebisacrylamide, 0.5% TEMED, 0.5% ammonium persulfate in PBS). The gels were allowed to settle for 5 min at RT, then incubated at 37°C for 30 min. The coverslips were then carefully removed from the gels by slight expansion of the gels in $dH_2O$, and the gels were immediately transferred into denaturation buffer and incubated at 95°C for 1 h. After denaturation, gels were expanded by washing in $dH_2O$ 3×20 min and subsequently shrunk in PBS for 30 min. For antibody staining, one-fourth of the gel was cut with a razor and incubated in primary antibody diluted in blocking buffer (2% BSA, 0.02% sodium azide in PBS) 6 h to overnight. The unbound primary antibody was removed by washing in PBS 3×20 min and the gels were subsequently incubated in secondary antibody diluted in blocking buffer 6 h to overnight. The unbound secondary antibody was removed by washed 3×20 min in PBS and the gels were incubated 90 min in 20 μg/mL fluorescent NHS ester (ATTO 594 or ATTO 488) in PBS. Finally, the gels were washed 3×20 min in PBS and expanded 2×30 min in $dH_2O$ prior to imaging.

### Confocal microscopy

For imaging expanded cytoskeletons, we used a Leica SP8 on an upright stand microscope DM6000 (Leica Microsystems, Mannheim, Germany), with a HCX Plan Apo CS 63X oil NA 1.40 objective. The lasers used were Diode 405 nm, OPSL 488, OPL 552 nm and Diode 638. The system was equipped with a conventional scanner (10 to 2,800 Hz) and four internal detectors (two conventional PMT and two hybrid detectors). The images were deconvolved in Huygens Pro and

processed with ImageJ and Fiji software [47,48]. The quantification in Fig 2G was obtained as follows. Eighteen slices of U-ExM confocal images were flattened with Fiji (Z-project SUM), and two areas were manually defined: a close area around the old FPC (excluding the BILBO1 labelling distal to the FPC), and the area from the old FPC to the basal bodies and including the old FPC and the MtQs. Fig 2G indicates the integrated fluorescence of BILBO1 at the old FPC as a percentage of the total fluorescence in the total area.

For imaging expanded whole cells, a Leica DMi8 with TCS SP8 confocal head (Leica Microsystems, Mannheim, Germany) equipped with the HC Plan APO CS2 63x oil NA 1.40 objective, a 20 mW 488 nm and a 20 mW 552 nm solid-state lasers, a standard scanner (1–1,800 Hz line frequency), and 5 detectors (2 × PMT and 3 × HyD), and a Leica DMi8 with STELLARIS 8 confocal head, with HC Plan APO 86x water immersion NA 1.20 objective, wide-range white light laser with pulse picker (WLL PP), conventional linear scanner (1–2,600 Hz line frequency), and five supersensitive hybrid detectors were used. The images were deconvolved using Huygens professional. To visualize the stages of FPC biogenesis, the data were segmented in Bitplane Imaris software by manual segmentation combined with intensity thresholding.

### Electron microscopy

A total of 50 mL of mid-log phase WT or 48-h BILBO1 RNAi-induced cells were harvested by centrifugation at 1,000*g* for 15 min. Block preparation and EM protocol were performed as in [41].

### Statistical analysis

Statistical analyses were performed using GraphPad Prism. Distance measurements in Fig 6E were obtained for four target protein combinations (Spef1-Tubulin as control, BILBO1-Tubulin, MORN1-Tubulin, BILBO1-BILBO2 with 18–21 measurements (1 or 2 measurements per image). The distribution of each dataset was assessed using the Shapiro–Wilk normality test. Since two of the four groups (Spef1-Tubulin, BILBO1-BILBO2) deviated significantly from a normal distribution ($p \leq 0.05$), a non-parametric Kruskal–Wallis test was applied to evaluate overall group differences. Statistical significance was defined as $p < 0.05$, and is indicated on the graph as follows: $p < 0.01$ (**), $p < 0.001$ (***), and non-significant comparison as "ns." In Fig 2G, following normality test, ANOVA test was performed with a statistical significance defined as $p \leq 0.05$.

### Supporting information

**S1 Fig. U-ExM immunofluorescence images showing that a sub-population of BILBO1 (yellow) and MORN1 (cyan) localize close to the centrin arm, which is indicated by $_{myc}$CAAP1 labelling (magenta).** Scale bars: 1 µm. (TIF)

**S2 Fig. NHS ester labelling of whole expanded cells. a** 3D rendering of two cells labelled with NHS ester (gray) and an antibody against BILBO1 (yellow). **b** A single Z-plane of the same data viewed through the orthoslicer. The arrow points to the flagellar pocket (FP), which is not visible in whole-cell rendering. **c** Segmentation of the data shown in (a) and (b). The cell surface (transparent gray) was segmented from the NHS ester signal by intensity thresholding. Microtubule-based structures at the flagellum base (solid gray) were segmented by manual masking combined with intensity thresholding. The FP (purple, arrow) was modelled by manual segmentation, and the BILBO1 signal (yellow) was segmented by intensity thresholding. **d** Maximum intensity projection of NHS ester labelling (inverted LUT). NHS ester labels organelles and structures within the cell, such as the nucleus (N), mitochondrion (M), glycosomes (G), the flagellum (Fg), and the basal bodies (BB). **e** A single Z-plane of the data in (d). Slicing through the volume reveals additional details, such as the shape of the flagellar pocket (FP). **f-i** Details of single Z-planes of the data represented in (d) and (e). **f,g** Details of the flagellar pocket. **h** Nucleus with visible nucleolus (arrow). **i** Detail of glycosomes (arrows). Scale bars: 10 µm in a-e, 5 µm in f-i. (TIF)

**S3 Fig. Colocalization of MORN1 and BILBO1.** Here, BILBO1 (yellow) is tagged with a C-terminal 3xHA-tag and labelled with an anti-HA antibody. MORN1 (cyan) is labelled with a MORN1 antibody. Scale bar: 2 µm physical distance (correspond to 0.43 µm after correction for the 4.6 fold expansion factor). **a**—Stage 1. **b**—Stage 2. **c**—Stage 4 during new flagellar pocket collar (pro-FPC) formation. **d**—Stage 4 with the interconnecting fibre (IF). (TIF)

**S1 Table. List of antibodies and NHS ester reagents used in this study.** This table provides detailed information on all primary and secondary antibodies, as well as NHS ester-based fluorescent reagents, used throughout the experiments. For each item, the table lists the target or label, host species (for antibodies), fluorophore, supplier, catalog number or reference, and dilution or concentration used. (PDF)

**S1 Data. Raw measurements underlying the quantitative analyses presented in Figs 2G, 3B, and 6E.** (XLSX)

## Acknowledgments

We thank the ProParacyto group members for stimulating discussions. We thank B. Morriswood for the anti-MORN1 antibody and critical reading of the manuscript, K. Ersfeld for the 9E10 anti-myc antibody, and S. Dean for the pPOTv7 vectors. We thank Martin Bablon (Freelance artist - Paris, France) for his help to design the scheme in Fig 8. We also acknowledge the Bordeaux Imaging Centre, a service unit of the CNRS-INSERM and Bordeaux University, a member of the national infrastructure France BioImaging supported by the French National Research Agency (ANR-10-INBS-04). The help of Christel Poujol, Magali Mondin, and Mónica Fernández Monreal is acknowledged. We also acknowledge the Light Microscopy Core Facility, IMG, Prague, Czech Republic, supported by MEYS – LM2023050, MEYS – CZ.02.1.01/0.0/0.0/18_046/0016045 and MEYS – CZ.02.01.01/00/23_015/0008205, for their support with the confocal imaging and image analysis presented herein.

## Author contributions

**Conceptualization:** Derrick Roy Robinson, Mélanie Bonhivers.

**Data curation:** Marie Zelená, Vladimir Varga, Mélanie Bonhivers.

**Formal analysis:** Marie Zelená, Elina Casas, Chloé Lambert, Denis Dacheux, Vladimir Varga, Derrick Roy Robinson, Mélanie Bonhivers.

**Funding acquisition:** Gang Dong, Vladimir Varga, Derrick Roy Robinson, Mélanie Bonhivers.

**Investigation:** Marie Zelená, Elina Casas, Chloé Lambert, Vladimir Varga, Derrick Roy Robinson, Mélanie Bonhivers.

**Methodology:** Marie Zelená, Elina Casas, Chloé Lambert, Nicolas Landrein, Vladimir Varga, Derrick Roy Robinson, Mélanie Bonhivers.

**Project administration:** Mélanie Bonhivers.

**Resources:** Mélanie Bonhivers.

**Supervision:** Vladimir Varga, Mélanie Bonhivers.

**Validation:** Gang Dong, Vladimir Varga, Derrick Roy Robinson, Mélanie Bonhivers.

**Visualization:** Marie Zelená, Elina Casas, Eloïse Bertiaux, Vladimir Varga, Derrick Roy Robinson, Mélanie Bonhivers.

**Writing—original draft:** Mélanie Bonhivers.

**Writing—review & editing:** Marie Zelená, Chloé Lambert, Denis Dacheux, Eloïse Bertiaux, Kim Ivan Abesamis, Gang Dong, Vladimir Varga, Derrick Roy Robinson, Mélanie Bonhivers.

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
