## [Editor Report · Decision Letter 0]

8 May 2025

Dear Dr Bonhivers,

Thank you for submitting your manuscript entitled "Flagellar pocket collar biogenesis: Cytoskeletal organization and novel structures in a unicellular pathogen" for consideration as a Research Article by PLOS Biology. Please accept my apologies for the delay in getting back to you as we consulted with an academic editor about your submission.

Your manuscript has now been evaluated by the PLOS Biology editorial staff, as well as by an academic editor with relevant expertise, and I am writing to let you know that we would like to send your submission out for external peer review.

Once your full submission is complete, your paper will undergo a series of checks in preparation for peer review. After your manuscript has passed the checks it will be sent out for review. To provide the metadata for your submission, please Login to Editorial Manager (https://www.editorialmanager.com/pbiology) within two working days, i.e. by May 10 2025 11:59PM.

Kind regards,

Richard

Richard Hodge, PhD

rhodge@plos.org

PLOS

---

## [Decision Letter · Decision Letter 1]

5 Jun 2025

Dear Dr Bonhivers,

Thank you for your patience while your manuscript "Flagellar pocket collar biogenesis: Cytoskeletal organization and novel structures in a unicellular pathogen" went through peer-review at PLOS Biology. Your manuscript has now been evaluated by the PLOS Biology editors, an Academic Editor with relevant expertise, and by three independent reviewers.

As you will see, the reviewers are all very positive about your study and think it provides a strong contribution to trypanosome cell biology. In light of the reviews, which you will find at the end of this email, we are pleased to offer you the opportunity to address the comments from the reviewers in a revision that we anticipate should not take you very long. We will then assess your revised manuscript and your response to the reviewers' comments with our Academic Editor aiming to avoid further rounds of peer-review, although we might need to consult with the reviewers, depending on the nature of the revisions.

We expect to receive your revised manuscript within 2 months. Please email us (plosbiology@plos.org) if you have any questions or concerns, or would like to request an extension.

**IMPORTANT - SUBMITTING YOUR REVISION**

*Resubmission Checklist*

*Published Peer Review*

*PLOS Data Policy*

*Blot and Gel Data Policy*

Best regards,

Richard

Richard Hodge, PhD

rhodge@plos.org

REVIEWS:

Reviewer #1: The trypanosome flagellar pocket with its pocket collar is a unique structure of Kinetoplastids, whose function, assembly and composition is still not fully understood. In this publication, the authors have analysed the division of the trypanosome flagellar pocket with all the attached features (hook complex, flagellar pocket collar, basal bodies, flagella). They employed UExM on detergent extracted cytoskeleton and whole cells, in combination with NHS total protein stain and a range of specific antibodies. The revisited the BILBO1 RNAi phenotype and could describe the function of BILBO1 more accurately.

This work constitutes an impressive bit of excellently-performed cell biology on a highly complex 3D structure. The authors describe the division in novel detail, and discover a few new sub-structures, such as the flagellar-pocket-collar-interconnecting fibre (a transient structure present during division). They show that the FPC remains tubulin-negative during the entire cell cycle. This work is an excellent addition to the trypanosome cell biology, and I have only a few minor comments.

* Introduction: could the authors state on how unique the flagellar pocket collar is, given that flagellar pockets appear to be present in many cells?

* Figure 6: The measurements appear to be based on 2-3 single measurements, it would benefit from a larger number of measurements and a bit of statistics.

* MORN1 and BILBO1 give rather similar stains, indicating (partial) colocalization, but the authors could not directly show this, due to technical issues. I do not know the nature of the technical issues, but if these are minor and could be solved (for example by creating a new cell line with a different epitope tag etc), it would be nice to see the direct comparison between these two proteins.

* Line 186: "With the exception of the G1-phase cells (where no nFPC is being formed) and the period when the FPC-IF is detected, the oFPC consistently accounts for approximately 40% of the total BILBO1 fluorescence, remaining close to 50% throughout the cell cycle. This indirectly suggests that nFPC biogenesis occurs de novo, rather than through redistribution of BILBO1 from the oFPC." I failed to understand this: If there would be redistribution of BILBO1 from the oFPC rather than de novo assembly, why would you expect different percentages?

* Given that this is a rather complex, 3-dimensional structure: would it be possible, to include a model of the pocket division to the discussion part, summarising all the major structures and findings? This would help the non-expert readers a lot.

* Line 126: Typo (expressing)

Reviewer #2: This paper describes a microscopy-based analysis of biogenesis of one of the key vital cytoskeletal structures in Trypanosoma brucei parasites, the hook complex/flagellar collar structure.

The paper is very single method-focused, but applies it very well, making use of the power of expansion microscopy. This is a powerful application, as these are highly structured structures which are on the edge of being resolved by conventional light microscopy, but well-resolved by expansion microscopy, maximising the value of this approach. Also, some of these structures, particularly the hook complex, are not readily visible by electron microscopy, making this a powerful approach for targeted consideration of known proteins. This comprehensively maps how this vital structure duplicates for cell division, providing insights into complex flagellar pocket morphogenesis.

A limitation is that expansion microscopy cannot distinguish between assembled material and material which is in transient structures or simply being transported. Flux of material will bias the visualised protein to assembled material, but this is a limitation. It was not always clear whether samples for particular experiments are cytoskeletons or whole cells, where material in transport is more likely to be extracted. Please try consistently indicate in the text and figures.

It is powerful that BILBO and MORN are antibody to native protein, avoids problems with assembly differences for tagged/untagged allele, or subtle defects due to a tag. However, more care should be taken describing the microtubule quartet, as it extends along the entire FAZ. The SPEF1 positive portion (and BILBO/MORN association) is only the proximal microtubule quartet, and that should be made clear whenever the microtubule quartet is discussed (given the abbreviation-heavy text, pMtQ for proximal microtubule quartet might be useful).

Much as the core data is qualitative. However, all figures would benefit from sample size indications. Making it clear from how many examples the selected example is representative of - both number of sample preparations and number of cells from which the example was selected. For example:

Figure 1: For each, representative example from how many. But, this matches tomography data, so less important to have high confidence, it meets expectations.

Figure 2: For each, representative example from how many. Particularly important to give confidence in the complex structures in stages 4, 5.

Figure 3: A, representative example from how many, B, sample size per data point.

Figure 4: For each, representative example from how many. Like Figure 3, confidence in the stage 4 and 5 examples.

Figure 6: Represantative of how many, and preferably add a mini bar chart of separation with a few data points per measurement.

Overall I found the text well written, easy to understand, well cited, etc. including the hard to describe complex three dimensional rearrangement in the results. I have a few minor comments about phrasing descriptions in the introduction and discussion.

Introduction

Line 75: Not well supported to describe the cortical cytoskeleton as a limitation. May well be a key advantage in different areas, and advantageous to concentrate all endo/exocytosis in one structure.

Line 81: May be worth noting similarity/analogy to feeding grooves and similar structures at the base of flagella in free living flagellates. Not vital though.

Line 86: Formally, the microtubule quartet is conserved, but there is diversity around this. The cytostome microtubules in cruzi and cytostome-containing monoxenous species like Paratrypanosoma, the additional pocket and cytoplasmic microtubules in Leishmania. More nuance would be good.

Line 88: Again, clarify about whether this is formally known for all trypanosomatid species, as the text structure implies that it is true/known.

Line 105: Clarify species where this evidence comes from, ie. T. brucei.

Line 110: State "in T. brucei".

Line 342: An extremely early event is Aurora Kinase association with the MtQ, worth quickly incorporating this point.

Line 353: Some more clarity about force generation here would be useful. The procyclic form flagella connector, vs. bloodstream form groove may provide a force.

Line 380: IFT is exceptionally specific to flagella, this would be very odd. Worth some consideration on likely trafficking direction of kinesins, although I realise this can be hard to predict. More generally, do you think there is any informative link with potential transport of material along the entire microtubule quartet, for supporting eg. FAZ assembly or coordinating ER attachment?

Line 401: Do I understand correctly that the new flagellar pocket collar does not have a presence in the early ridge between the dividing pockets? Possibly worth reversing this statement to say that your results show that it is unlikely that collar or hook complex rearrangements are fully responsible for early ridge formation.

Reviewer #3 (Markus Engstler, signs review):

General

This is an excellent manuscript that presents a comprehensive and technically rigorous analysis of flagellar pocket collar biogenesis in Trypanosoma brucei, using advanced ultrastructure expansion microscopy.

The identification and characterization of two previously undescribed structures, the pro-flagellar pocket collar (proFPC) and the FPC-interconnecting fibre (FPC-IF), represent a major advance in our understanding of how new FPCs are assembled de novo during the cell cycle. The application of ultrastructure expansion microscopy is particularly powerful here, enabling unprecedented resolution of spatial and temporal dynamics of FPC, HC, and MtQ components. These discoveries revise and extend existing models of FPC biogenesis and may also provide broader insights relevant to ciliary pocket formation in other eukaryotic systems.

Major

1. Mechanistic interpretation of proFPC and FPC-IF formation

While the structural characterization of these elements is excellent, the manuscript could benefit from a more detailed discussion on potential molecular mechanisms underlying their formation. The authors briefly mention motor proteins such as TbKINX1B, but do not elaborate on possible regulatory or recruitment pathways. A more speculative discussion here would add value without requiring additional experiments.

2. Functional implications and broader context

The authors are encouraged to elaborate further on the potential biological role of the transient FPC-IF and the significance of its rapid disappearance. How might these structures contribute to flagellar pocket dynamics, immune evasion, or membrane remodeling? Even in the absence of direct experimental data, this would help position the work more broadly within parasite cell biology.

3. Clarity and terminology

Consider standardizing terminology when referring to old/new structures (e.g., oFPC/nFPC vs. proFPC). In some instances, the transitions between stages and structures could be clearer. A table or expanded legend in Figure 8 summarizing each step in the proposed biogenesis pathway would aid readability.

Minor

Line 96: "…serving as a the structural backbone…"

Abstract: Consider condensing the background portion to strengthen the focus on findings.

A useful model figure, but the legend should be expanded to clarify each structure and its temporal context.

Figure legends: Ensure all abbreviations (e.g., FC, BB, FAZ) are defined on first use in figure legends, not just the main text.

Data and methods

The methodological descriptions are thorough and appropriate. The use of detergent-extracted and whole-cell U-ExM, coupled with well-validated antibodies, is convincing. Data availability is clearly addressed.

While the morphological data and imaging results are compelling, the manuscript would benefit from greater transparency and rigor in the reporting of quantitative analyses.

Specifically:

Please clarify the number of biological replicates and individual cells measured for each quantitative analysis (e.g., fluorescence intensity in Figure 2G, distances in Figure 3B).

For datasets involving continuous measurements (e.g., nBB-nFPC distances), consider reporting descriptive statistics such as mean ± SD or using graphical formats (e.g., box plots) that reflect data distribution.

Indicate whether quantification was performed in a blinded fashion and whether statistical comparisons (e.g., tests for significance between stages) were applied where appropriate.

While these additions are not critical for validating the overall conclusions, they would significantly enhance the statistical rigor and reproducibility of the study.

Conclusion

This is a highly original and well-executed study that makes a strong contribution to the field. It is suitable for publication in PLOS Biology pending minor revisions.

---

## [Editor Report · Decision Letter 2]

11 Sep 2025

Dear Dr Bonhivers,

Thank you for your patience while we considered your revised manuscript "Flagellar pocket collar biogenesis: Cytoskeletal organization and novel structures in a unicellular pathogen" for publication as a Research Article at PLOS Biology. This revised version of your manuscript has been evaluated by the PLOS Biology editors and the Academic Editor.

Based on our Academic Editor's assessment of your revision, I am pleased to say that we are likely to accept this manuscript for publication, provided you address the following data and other policy-related requests that I have provided below (A-D):

(A) We routinely suggest changes to titles to ensure maximum accessibility for a broad, non-specialist readership. In this case, we would suggest a minor edit to the title, as follows. Please ensure you change both the manuscript file and the online submission system, as they need to match for final acceptance:

"Visualization of Trypanosoma brucei flagellar pocket collar biogenesis identifies two new cytoskeletal structures"

(B) You may be aware of the PLOS Data Policy, which requires that all data be made available without restriction: http://journals.plos.org/plosbiology/s/data-availability. For more information, please also see this editorial: http://dx.doi.org/10.1371/journal.pbio.1001797

-Supplementary files (e.g., excel). Please ensure that all data files are uploaded as 'Supporting Information' and are invariably referred to (in the manuscript, figure legends, and the Description field when uploading your files) using the following format verbatim: S1 Data, S2 Data, etc. Multiple panels of a single or even several figures can be included as multiple sheets in one excel file that is saved using exactly the following convention: S1_Data.xlsx (using an underscore).

-Deposition in a publicly available repository. Please also provide the accession code or a reviewer link so that we may view your data before publication.

Figure 3B, 6A-E

(C) Please also ensure that each of the relevant figure legends in your manuscript include information on *WHERE THE UNDERLYING DATA CAN BE FOUND*, and ensure your supplemental data file/s has a legend.

(D) Per journal policy, if you have generated any custom code during the course of this investigation, please make it available without restrictions. Please ensure that the code is sufficiently well documented and reusable, and that your Data Statement in the Editorial Manager submission system accurately describes where your code can be found.

We expect to receive your revised manuscript within two weeks.

*Published Peer Review History*

*Press*

Best wishes,

Richard

Richard Hodge, PhD

rhodge@plos.org

PLOS

---

## [Editor Report · Decision Letter 3]

19 Sep 2025

Dear Dr Bonhivers,

On behalf of my colleagues and the Academic Editor, André Schneider, I am pleased to say that we can accept your manuscript for publication, provided you address any remaining formatting and reporting issues. These will be detailed in an email you should receive within 2-3 business days from our colleagues in the journal operations team; no action is required from you until then. Please note that we will not be able to formally accept your manuscript and schedule it for publication until you have completed any requested changes.

PRESS

Best wishes,

Richard 

Richard Hodge, PhD

rhodge@plos.org

PLOS
